# Deep flanking sequence engineering for efficient promoter design using DeepSEED

Pengcheng Zhang [1,3], Haochen Wang[1,3], Hanwen Xu[1,3], Lei Wei[1], Liyang Liu[1], Zhirui Hu[2] & Xiaowo Wang [1] ✉

Designing promoters with desirable properties is essential in synthetic biology. Human experts are skilled at identifying strong explicit patterns in small samples, while deep learning models excel at detecting implicit weak patterns in large datasets. Biologists have described the sequence patterns of promoters via transcription factor binding sites (TFBSs). However, the flanking sequences of cis-regulatory elements, have long been overlooked and often arbitrarily decided in promoter design. To address this limitation, we introduce DeepSEED, an AI-aided framework that efficiently designs synthetic promoters by combining expert knowledge with deep learning techniques. DeepSEED has demonstrated success in improving the properties of *Escherichia coli* constitutive, IPTG-inducible, and mammalian cell doxycycline (Dox)-inducible promoters. Furthermore, our results show that DeepSEED captures the implicit features in flanking sequences, such as k-mer frequencies and DNA shape features, which are crucial for determining promoter properties.

Promoters are core genetic elements that regulate gene expression[1]. Designing synthetic promoters with desirable properties to precisely control gene expression is a requisite for biosynthetic engineering and gene therapy[2,3]. Strong or inducible promoters are indispensable for achieving high transgene expression and maximizing the potency of the treatment[4–6]. It has been believed that the properties of a promoter are mostly determined by cis-regulatory elements[7,8], i.e., transcription factor binding sites (TFBSs), such as the −10/−35 elements in prokaryotes and the TATA-box in eukaryotes[9,10]. The sequence preference of TFBSs is commonly summarized as motifs and represented by position weight matrices (PWMs). As the sequences and functions of TFBSs are usually well known, most researchers design new promoters by manipulating the combinations and arrangements of TFBS motifs[11,12]. However, recent evidence has highlighted that TFBS flanking sequences also significantly influence promoter properties[13–15]. The roles of the flanking sequence around TFBSs encompass various aspects, including the influence of physicochemical properties such as DNA shapes[16,17], specific flanking sequence preferences by certain TFBSs[12,14,18], and the presence of potential low-affinity binding sites in flanking regions that enhance TF binding[15,19–21]. It is difficult to summarize these features into explicit promoter design rules; thus, the optimization of flanking sequences is largely unexplored in current promoter design approaches.

Recently, deep learning models have shown great potential in both eukaryotic and prokaryotic promoter engineering[10,22–24]. By capturing shared sequence patterns from a large number of natural promoters, these models have been able to generate numerous de novo constitutive promoters with high diversity. However, these data-driven deep learning models cannot address the demands of designing promoters with specific properties such as inducible promoters or tissue-specific promoters, as in nature there exist only a handful of promoters or even just a single promoter with the desired property for use in training[25–27]. Consequently, most synthetic promoters used in practice were designed through expert prior knowledge. For example, researchers may put TFBSs with related functions into a backbone sequence and then optimize the sequence with mutagenesis[12,27,28]. Such approaches lack instructive schemes for optimizing flanking sequences that also affect promoter properties, usually resulting in poorer-than-expected outcomes or painstaking trial-and-error tests.

[1]Ministry of Education Key Laboratory of Bioinformatics; Center for Synthetic and Systems Biology; Bioinformatics Division, Beijing National Research Center for Information Science and Technology; Department of Automation, Tsinghua University, Beijing, China. [2]Center for Statistical Science, Tsinghua University, Beijing, China. [3]These authors contributed equally: Pengcheng Zhang, Haochen Wang, Hanwen Xu. ✉e-mail: xwwang@tsinghua.edu.cn

Here, we propose DeepSEED (Deep learning-based flanking Sequence Engineering for Efficient promoter Design), an AI-aided flanking sequence optimization method for synthetic promoter design. DeepSEED aims to integrate expert knowledge with the power of data-driven models to facilitate efficient promoter design. Deep-SEED is composed of two deep learning models: a conditional generative adversarial network (cGAN)[29,30] that generates flanking sequences based on preset sequence elements, and a DenseNet-LSTM-based model (where LSTM denotes "long short-term memory") that predicts promoter properties. To design a synthetic promoter with desired properties, users can−based on their prior knowledge−first input any number of sequence elements (e.g., TFBSs) of interest at any position as a 'seed', and then DeepSEED will generate the flanking sequences based on the 'seed' to fit into the implicit patterns of promoters. Subsequently, to evaluate the significance of flanking sequences, we investigated distinct patterns of influence from the flanking regions in the functional *E. coli* promoters using the predictor model and saliency maps[31]. Additionally, we employed t-distributed stochastic neighbor embedding (t-SNE) to further analyze and confirm the relationship between promoter activity and DNA shape features in the flanking sequences. Next, we applied DeepSEED to three different promoter design tasks: prokaryotic constitutive promoters, prokaryotic IPTG-inducible promoters, and eukaryotic doxycycline-inducible promoters. In all three cases, DeepSEED showed significant improvement in achieving desired promoter properties with high success rates by optimizing flanking sequences. The synthetic promoters generated by DeepSEED showed high sequence diversity while preserving key features, such as k-mer frequencies and DNA shape features. These synthetic promoters demonstrated low sequence similarity to the natural genomic and comparable edit distance to randomly flanking sequences. Overall, these results underscored the crucial role of flanking sequences in promoter activity and suggested DeepSEED as a powerful AI-aided tool for designing synthetic promoters.

## Results

### Overview of the AI-aided flanking sequence optimization method DeepSEED

DeepSEED constructed a probabilistic view to unify the expert knowledge to define the 'seed' of promoters and deep learning methods to fill up the flanking sequences that match the 'seed' to improve the promoter performance. The promoter design problem can be formulated in probabilistic terms as maximizing the joint probability of the promoter sequence $s$ and the target property $T$, i.e.,

$$\max_s P(s,T) \tag{1}$$

The promoter sequence $s$ is divided into two parts: 'seed' sequences $m$ derived from expert knowledge and flanking regions of 'seed' $f$. By applying the chain rule, the following is obtained:

$$P(s,T) = P(m,f,T) \propto P(m|T)P(f|m,T) \tag{2}$$

The formula suggests that the maximization of the probability could be achieved in two stages (Fig. 1a). The first term $P(m|T)$ indicates the process of assigning $m$ compatible with the target property according to expert knowledge (Stage I, Expert Knowledge Integration). The second term $P(f|m,T)$ represents optimizing flanking regions $f$ conditioned on the 'seed' sequences $m$ and the property $T$ (Stage II, Sequence Optimization, see "Promoter design approach" in "Methods").

Expert Knowledge Integration: In Stage I, the maximization of the first term $P(m|T)$ is achieved by selecting 'seed' based on expert knowledge that has been proven to be essential for achieving the target property $T$. In the next stage, we assume that 'seed' sequences are fixed, defined as $m^*$.

Sequence Optimization: Given $m^*$ determined in Stage I, Stage II maximizes $P(f|m^*,T)$. According to the Bayes rule,

$$P(f|m^*,T) \propto P(f|m^*)P(T|f,m^*) \tag{3}$$

where the first term, $P(f|m^*)$, is related to the flanking sequence generation problem, in which one seeks the compatible flanking sequence $f$ based on $m^*$. The second term, $P(T|f,m^*)$, is the probability of a promoter property $T$ for the given sequence $s = (f,m^*)$.

A cGAN-based flanking sequence generation model was adopted in DeepSEED to estimate $P(f|m^*)$ (see "Objective function" in "Methods"). Attention-based layers were applied because of their ability to capture the widespread long-range interactions in regulatory codes. A DenseNet-LSTM-based predictor was trained to evaluate the properties of input promoters, i.e., $P(T|f,m^*)$ (Fig. 1b, Supplementary Fig. 1). The genetic algorithm (GA) combining the cGAN generator and the predictor was applied to maximize the probability of $P(f|m^*,T)$ to design synthetic promoters with target properties (Fig. 1c, Supplementary Fig. 2a). These well-designed network structures and training strategies help improve the performance of DeepSEED compared to our previous methods[22] (Supplementary Notes). In the following sections, we will demonstrate that DeepSEED can learn the implicit pattern in flanking sequences and can design high-activity promoters based on expert knowledge (Fig. 1d).

### DeepSEED captures essential features of natural promoter sequences

We first investigated the importance of flanking sequences in predicting promoter activity in silico. We predicted the expression level of 2000 functional *E. coli* promoters by the predictor model and clustered their saliency maps[31] (see "Saliency map analysis" in "Methods"). As shown in Fig. 2a and Supplementary Fig. 3, we found that the −10 and −35 elements strongly influence promoter activity, as reported by previous studies[32]. However, the influence of flanking regions is also important. Each cluster showed distinct patterns in flanking regions on the saliency maps, and the potential relationship of flanking sequences to promoter activity is difficult to describe explicitly.

We then evaluated whether the DeepSEED architecture can capture the implicit patterns of flanking sequences at specific and entire promoter regions. For *E. coli* promoters, we observed that the k-mer frequency between natural and DeepSEED-designed sequences correlated well (k = 4 to 6) across entire promoter regions (Fig. 2b, Supplementary Fig. 4a). We further investigated the k-mer frequencies at the distal and proximal ends of the promoter, as they exhibit different k-mer patterns in natural sequences (Supplementary Fig. 4b), and the results showed that DeepSEED successfully captured the k-mer features on both ends (Fig. 2b). The k-mer frequency in designed Dox-inducible promoters with preserved tetO sequences also showed a high correlation with natural sequences (Fig. 2c, Supplementary Fig. 4c).

DNA shapes are physicochemical features of DNA sequences that contain structural information, such as the minor groove width (MGW), roll, propeller twist (ProT), and helix twist (HelT)[33]. Appropriate DNA shapes near TFBSs could enable TF binding and thus influence transcriptional activity[34–36]. We first examined the four kinds of DNA shape features on the left side of two widespread −10 elements, 'TATAAT' and 'TATAAA', in *E. coli* and found that DeepSEED-designed promoters showed patterns more similar to those of natural promoters than to random sequences at specific locations of the promoter region (Fig. 2d). We further embedded DNA shape features of whole promoter sequences by t-SNE (see "DNA shape analysis" in "Methods") and found that *E. coli* promoters in the training dataset can be divided into two groups in the embedding space related to their transcriptional activities, implying the strong dependence between DNA shapes and promoter activities (Fig. 2e). Moreover, we noticed that DeepSEED optimization can successfully move a promoter

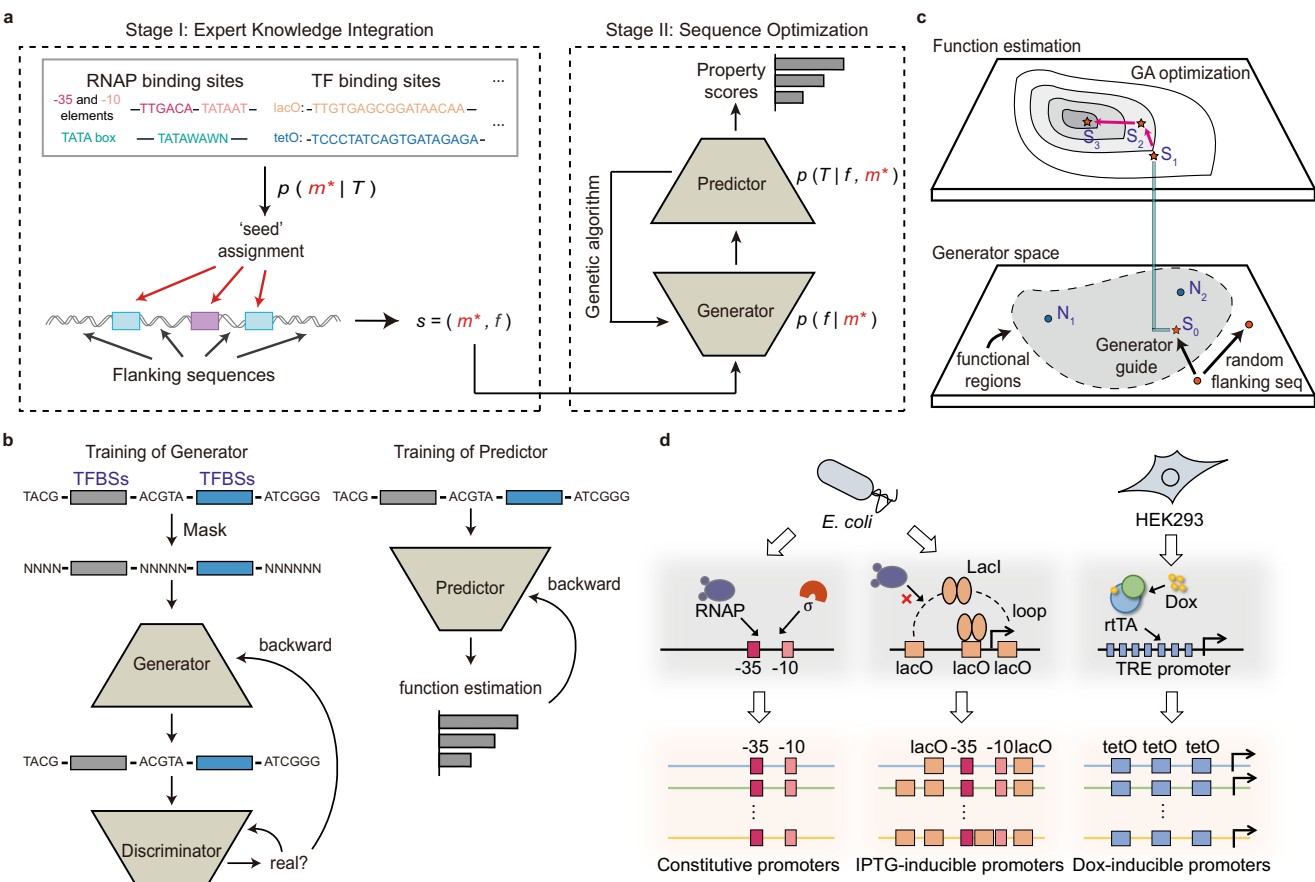

**Fig. 1 | An overview of the DeepSEED approach. a** The learning procedure of DeepSEED. The model was able to automatically design the promoter sequences by means of the two stages of the learning process. **b** Training process of the generator and predictor. The details of the sequence construction in the three promoter design tasks are shown in Supplementary Fig. 6. **c** Sequence generation process in the generator and genetic algorithm. The functional regions represent the functional promoter sequence space learned by the generator. $N_1$ and $N_2$ represent the natural sequences from the dataset. $S_0$, $S_1$, $S_2$, and $S_3$ represent the sequences in the optimization process. **d** The promoter design tasks in this work. The protein binding sites, as prior expert knowledge, were used to design constitutive promoters, IPTG-inducible promoters, and Dox-inducible promoters.

sequence from the low-activity region to the high-activity region in the embedding space.

We also investigated the promoter sequences from the perspective of the DNA semantic sequence space (see "Semantic sequence space" in "Methods"). Compared with promoter sequences of other species in the training dataset, the DeepSEED-designed promoter sequences and natural *E. coli* promoters are co-located in the same low-dimensional space (Supplementary Fig. 5a). Furthermore, we demonstrated that DeepSEED-designed promoters showed similar diversity to natural sequences by comparing the editing distance distributions (Supplementary Fig. 5b, see "Sequence similar diversity" in "Methods"), indicating that DeepSEED has the ability to generate brand-new promoters rather than simply modifying natural promoters. All the above results suggested that DeepSEED was able to capture essential features of natural promoter sequences and direct the model to generate de novo high-activity promoters.

### Design of constitutive promoters in *E. coli*

We first used DeepSEED to design constitutive promoters in *E. coli*. We fixed the sequence and position of −10/−35 elements as 'seed' sequences (Fig. 3a). DeepSEED was trained on promoters from the massively parallel reporter assay (MPRA) dataset[37] to optimize flanking sequences to improve the activity of promoters (Supplementary Fig. 6a). Compared with natural *E. coli* promoters containing the same −10 and −35 elements, DeepSEED-designed sequences were predicted to have higher expression levels in silico (Supplementary Fig. 2b).

Then, we verified the properties of DeepSEED-designed sequences in vivo. Three widely used constitutive promoters adopted from the Internationally Genetically Engineered Machine (iGEM) parts registry (BBa_J23119, BBa_J23118, and BBa_J23114, http://parts.igem.org/Promoters/Catalog/Constitutive) with different −10 and −35 elements were selected as the initial sequences to be optimized. The three promoters showed high, medium, and low expression levels in vivo (Supplementary Fig. 7). For each initial sequence, we kept the 'seed' sequences and designed 16 new sequences with DeepSEED (Supplementary Notes). For the sake of comparison, we also generated two control groups (Fig. 3b): the Control-1 group using random sequences to extend the initial promoter sequences to 165 bp, the same length as promoters in the training dataset, to prevent the effect of length on promoter activity; and the Control-2 group in which the 'seed' sequences were maintained while randomizing all the other regions (Supplementary Data 1). We analyzed the edit distance between the initial sequences and the corresponding experimental sequences of equal length. The DeepSEED-generated promoters showed sequence differences of $51.39 \pm 4.66\%$, $45.14 \pm 3.63\%$, and $46.06 \pm 5.23\%$ compared to the J23114, J23118, and J23119 initial sequences, respectively. These differences were comparable to the randomly generated sequences observed in the Control-2 group ($52.31 \pm 6.33\%$, $51.39 \pm 3.55\%$, and $50.93 \pm 6.42\%$, respectively; Supplementary Data 3, see "Edit distance analysis" in "Methods"). Moreover, the results of the BLAST search on DeepSEED-generated constitutive promoters showed similar levels to random sequences, and lower similarity with the

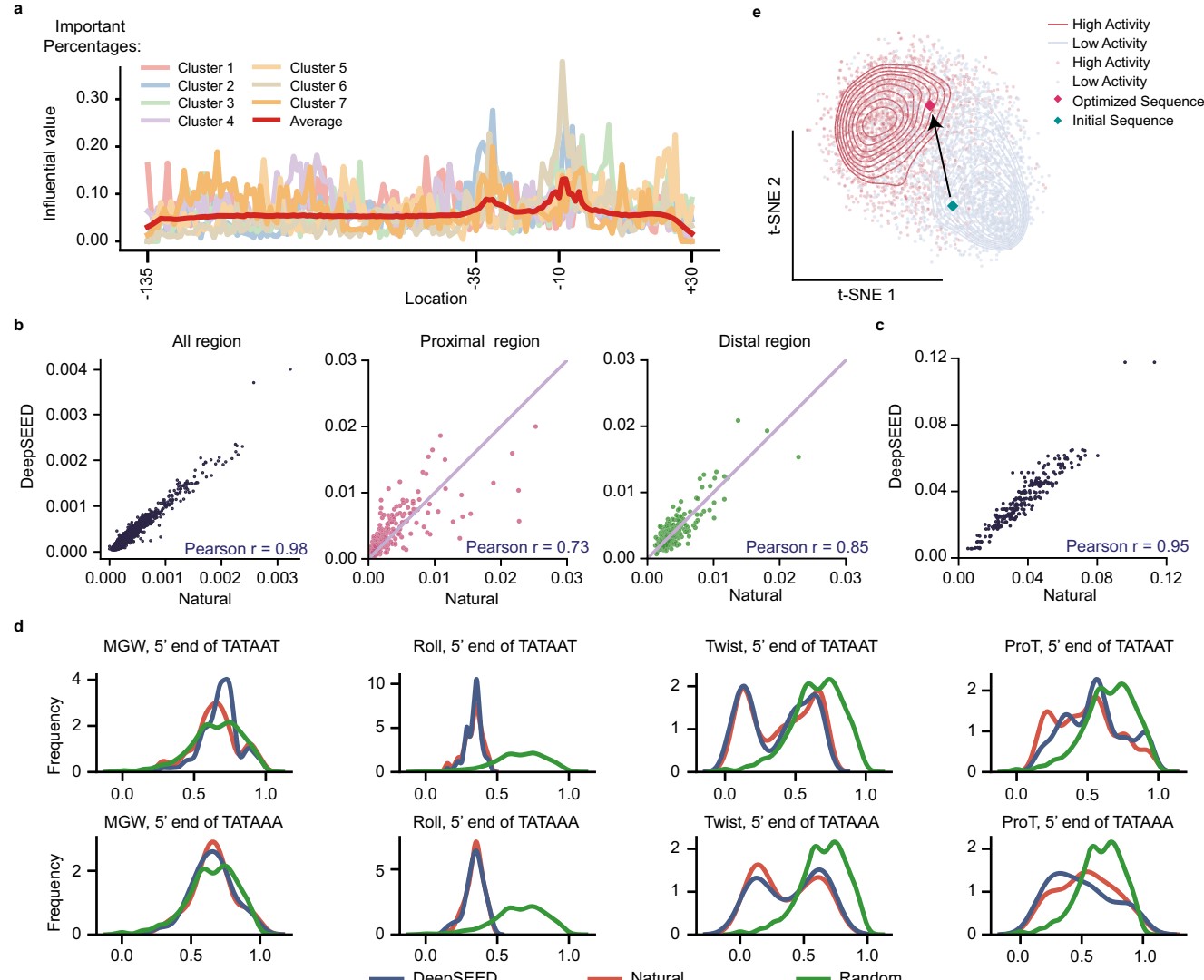

**Fig. 2 | DeepSEED captures implicit patterns in flanking sequences. a** The gene expression influential values of each location in the natural promoter sequences. The x-axis represents the nucleotide location corresponding to the transcription start site (TSS). The details of each cluster are shown in Supplementary Fig. 3. **b**, **c** Scatter plot showing 4-mer frequency between natural and DeepSEED-generated *E. coli* sequences in entire, distal, and proximal promoter regions (**b**) and between natural and DeepSEED-generated Dox-inducible sequences in entire

promoter regions (**c**), in which each dot represents one specific 4-mer. **d** DNA shapes (MGW, Roll, Twist, ProT) of natural and DeepSEED-generated sequences on the 5' end of two types −10 sequences, 'TATAAT' and 'TATAAA'. **e** *E. coli* promoters in the DNA shape embedding space. Dots in blue and red represent natural promoters with low and high activity. Diamonds in green and red represent a selected database promoter and the corresponding flanking sequence optimized promoter in silico. Source data are provided as a Source Data file.

natural *E. coli* genome than the promoters designed by Alper et al.[38] and most constitutive promoters from iGEM BioBrick standard parts[39] (Supplementary Fig. 8a, see "BLAST search" in "Methods").

We measured the constructed promoter activities in the LB medium by the fluorescence intensity of downstream sfGFP. As shown in Fig. 3c and Supplementary Fig. 9a, the promoter activities of the Control-1 group were comparable to those of the initial promoters. The Control-2 group showed a higher variance in promoter activities due to the introduction of more random sequences. The high variability in the activities of random flanking sequences in the Control-1 and Control-2 groups also indicated the importance of flanking sequences for promoter activity. After DeepSEED optimization, the activities of synthetic promoters in the DeepSEED group exhibited great improvement, increasing by an average of 1.42-, 4.11- and 33.43-fold compared with the Control-2 group based on the initial promoters BBa_J23119, BBa_J23118, and BBa_J23114, respectively. We compared the DeepSEED results with our previous whole sequence generation method[22] and found that DeepSEED showed an average 6.73-fold increase in

promoter activity (Fig. 3d, Supplementary Fig. 9b), suggesting that compared to learning only the general patterns from training samples, introducing expert knowledge such as the consensus motif in the promoter region may help the model generate promoters with better performance. In addition, we evaluated the activity of 21 synthetic promoters (Supplementary Data 1) using the reporter gene *mrfp*, and the results exhibited a strong correlation with the measurements using *sfgfp* (Pearson r = 0.83, Supplementary Fig. 10a). We further assessed the robustness of the DeepSEED-designed constitutive promoters by testing a subset of them in two additional types of culture medium: M9 and EZ-rich (Supplementary Fig. 10b, c, Supplementary Data 1). Remarkably, these synthetic promoters with different initial −10 and −35 motifs consistently displayed high activity levels in diverse culture conditions.

To exclude the possibility that the increase in promoter activity is due to the generation of a second promoter rather than to flanking sequence optimization, we analyzed the DeepSEED-designed promoters with motif-finding algorithms to evaluate the existence of

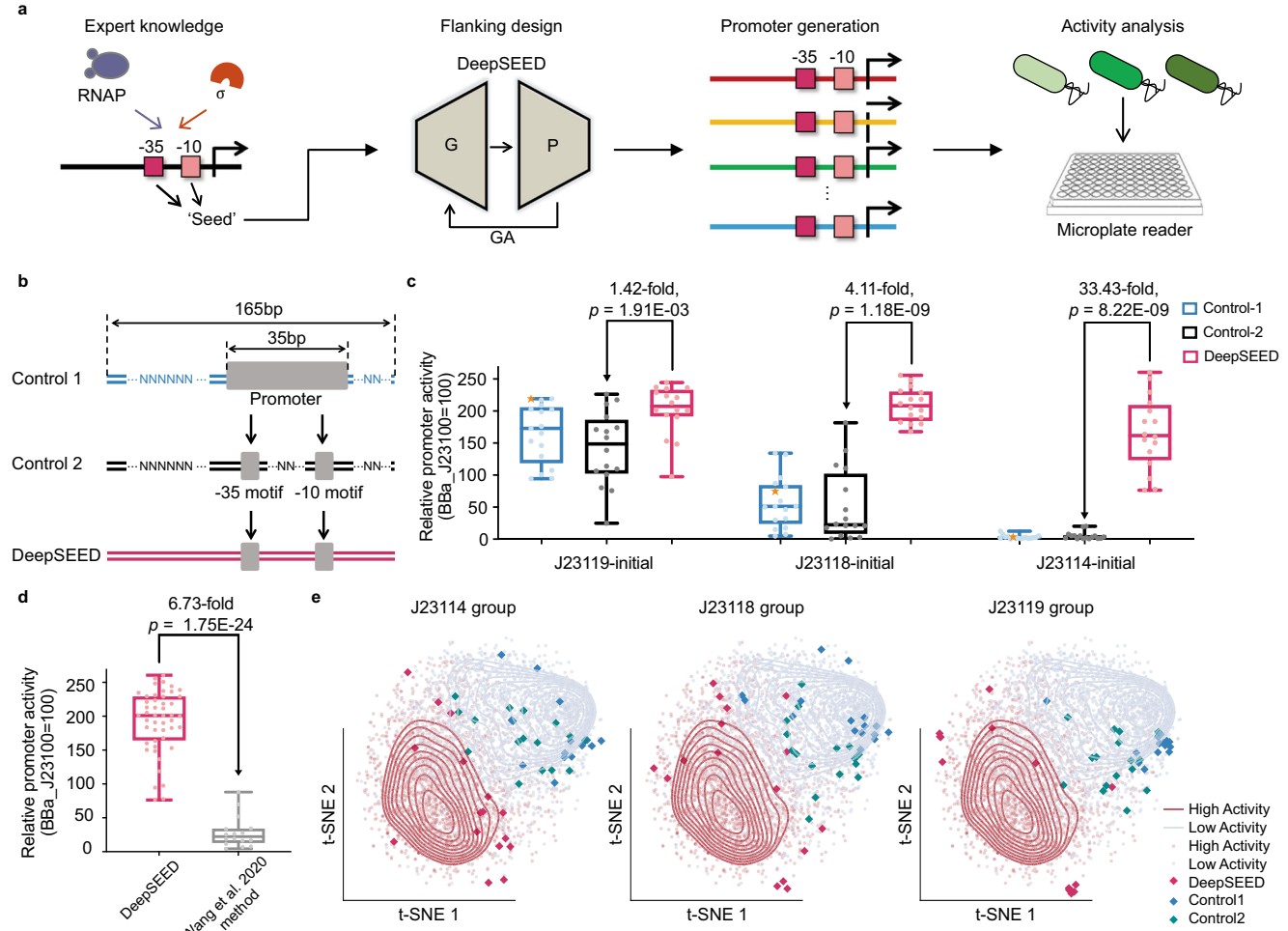

**Fig. 3 | The design of constitutive promoters in *E. coli*. a** The design process of the constitutive promoter in *E. coli*. **b** Schematic representation of the control sequences and DeepSEED-designed sequences tested in vivo. All promoters were 165 bp in length; Control-1 randomized the flanking sequence of the short J23 promoters; Control-2 preserved the −10/−35 elements as the DeepSEED group while randomizing other sequences; the flanking sequence of the DeepSEED group was generated by our method. All groups contained 16 promoters. **c** The boxplots show the promoter activity of Control-1, Control-2 and DeepSEED promoters. Orange stars in the three Control-1 groups represent the short 35 bp initial promoter. **d** Promoter activities generated by DeepSEED (*n* = 48) and the whole sequence generation methods from Wang et al. 2020[22] (*n* = 16) are shown. In **c**, **d**, the *p*-values were determined by a two-tailed unpaired Welch's *t*-test. The difference in average promoter activity between the DeepSEED group and the Control-2 or Wang et al. 2020 method group is also shown. The promoter activities were normalized by strong constitutive promoter J23100 from the iGEM parts registry. Each dot represents the average of three biological replicates. Box plots indicate the median (middle line), 25th, 75th percentile (box) and minimum and maximum values (whiskers). **e** Three promoter groups in the DNA shape embedding space. Dots in blue and red represent natural promoters with low and high activity. Red, blue, and green diamonds represent the validation constitutive promoters in vivo. Source data are provided as a Source Data file.

the alternative promoter within the flanking sequence (see "Second promoter finding" in "Methods"). We found that the signals of 'promoter structure' motifs in flanking sequence regions were almost always lower than those in the original regions (Supplementary Fig. 11). Although some increasing signals in flanking regions appeared, they were still much lower than those in the strongest promoter BBa_J23119. The results indicated that DeepSEED did not generate alternative high-activity promoter structures in flanking regions.

We plotted all the promoters and control samples in the DNA shape embedding space (Fig. 3e). Most of the Control-1 and Control-2 samples were located in low promoter activity regions, which indicated that random sequences in flanking regions failed to configure the DNA shape for higher transcriptional activity. In contrast, Deep-SEED optimized the flanking regions and generated highly expressed DNA shape features. These results indicated that the improvement of DeepSEED-designed promoters could partly be explained by the optimization of DNA shape features in flanking regions.

## Design of IPTG-inducible promoters in *E. coli*

In bacteria, the promoter contains not only the −10 and −35 elements but also regulatory sequences that could be bound by transcription factors to activate or repress gene expression, as with the lac promoter in the typical lac operon in *E. coli*[40]. The activity of the lac promoter is repressed when the LacI protein binds to lacO sites, and the repression can be relieved in the presence of lactose or its analogs, such as IPTG (Fig. 1d). The lac promoter is widely used in synthetic biology as an inducible promoter. We aimed to use DeepSEED to design new IPTG-inducible promoters based on the LacI-DNA interaction pattern as expert knowledge (Fig. 4a). Here, DeepSEED was trained on promoters from the MPRA dataset[37] to optimize flanking sequences to increase transcription under induced conditions (Supplementary Fig. 6b).

The number and position of the lacO sites are crucial for promoter performance. Appropriate spacing between different lacO sites may help to form a repression loop and further reduce leaky expression[25,41]. Here, we adopted two, three, or four lacO sites at suitable positions and spacing for LacI protein binding and loop formation[25] to design

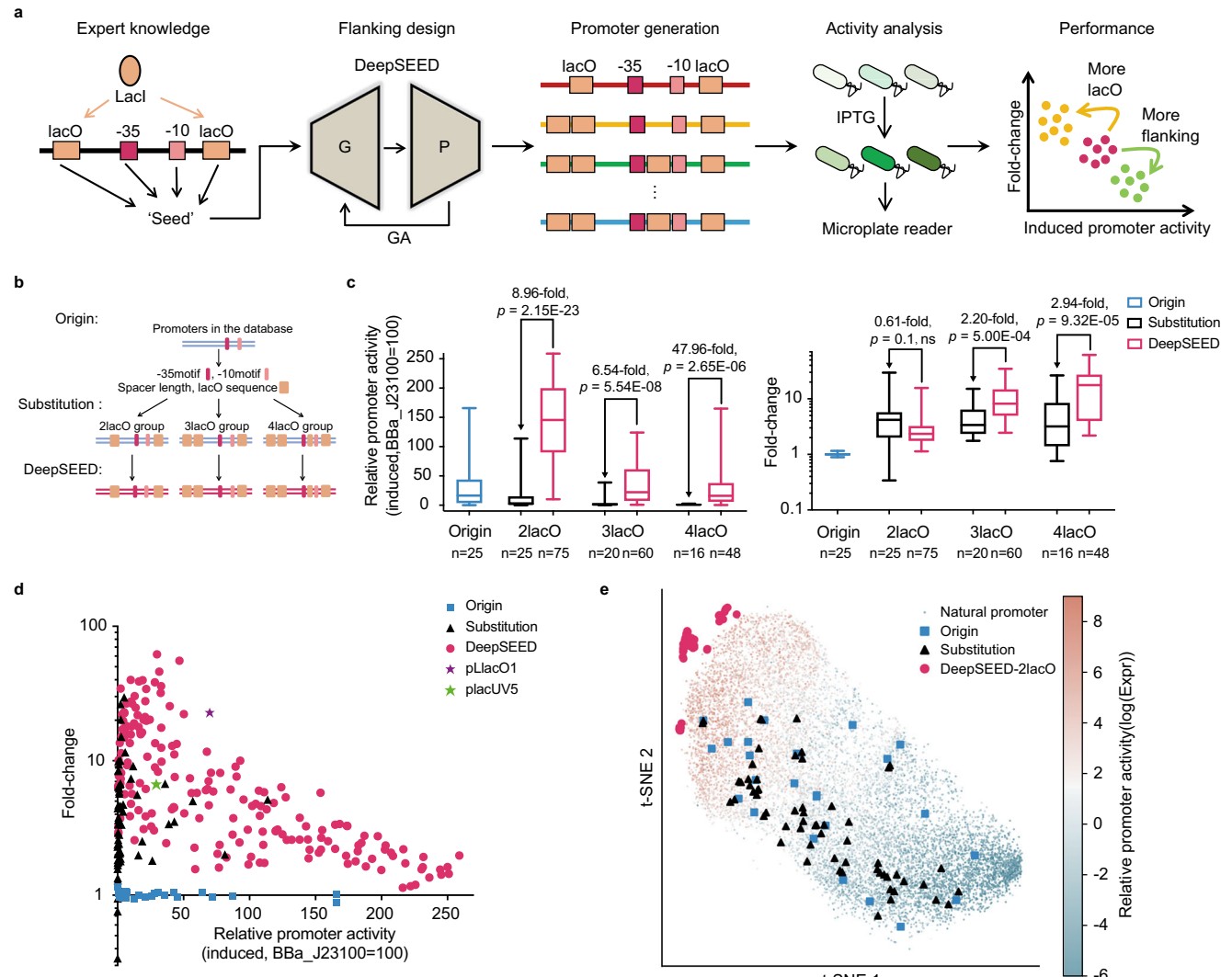

**Fig. 4 | The design of IPTG-inducible promoters in *E. coli*. a** The design process of the IPTG-inducible promoter in *E. coli*. **b** Schematic representation of the origin sequences, the substitution sequences, and the DeepSEED-designed sequences tested in vivo. We randomly chose the initial promoters from the dataset as the backbone. The −35, −10 elements, spacer length, and the number of lacO sequences as the 'seed'. The substitution group replaced the corresponding position sequence of the backbone with lacO sites, and the DeepSEED group optimized the flanking sequence of the promoters in the substitution group. **c** Boxplot showing induced activity and fold-change of the **b** promoters in vivo. The *p*-values were determined by a two-tailed unpaired Welch's *t*-test, where ns represents not significant. The difference in average induced activity and fold-change of the DeepSEED group and the substitution group are also shown. The promoter activities were normalized by strong constitutive promoter J23100 from the iGEM parts registry. Box plots indicate the median (middle line), 25th, 75th percentile (box) and minimum and maximum values (whiskers). **d** Two-dimensional images of the induced promoter activity and fold change of the promoters in **b**. Two wild-used promoters, pLlacO1 and placUV5, are also shown. Each dot represents the average of three biological replicates. **e** Promoters in the embedding spaces. Each dot represents one natural promoter with certain activities in the dataset. Validation promoters in **b** are also shown. Source data are provided as a Source Data file.

promoters (Supplementary Fig. 12). We randomly chose 25 constitutive promoters from the training samples as backbone sequences. We identified −10 and −35 elements and their spacer lengths for each sequence and adopted these features as well as the number and position of lacO sites as expert knowledge for the generative model (Fig. 4b). For each backbone sequence, we selected three sequences that were predicted to have high activity for experimental validation (Supplementary Notes). We compared the edit distance between the designed and template sequences. The 2-lacO, 3-lacO, and 4-lacO DeepSEED-generated promoters showed differences of 58.33 ± 4.08%, 60.11 ± 4.00%, and 61.84 ± 5.39%, respectively, which were similar to the randomly generated sequences (62.82 ± 3.59%, 63.69 ± 4.37%, and 64.58 ± 4.40%; Supplementary Data 3, see "Edit distance analysis" in "Methods"). These DeepSEED-generated promoters also exhibited low similarity with the natural *E. coli* genome (Supplementary Fig. 8a; see

"BLAST search" in "Methods"). In addition, we directly substituted the corresponding sequence with lacO sites on the backbone sequence as the substitution control group and measured their activities in vivo. The detailed sequences of promoters are provided in Supplementary Data 1.

We found that after the direct substitution, sequences showed different degrees of activity loss under the induced condition (Fig. 4c, Supplementary Fig. 13-15). The average induced expression level was reduced by 52.8%, 80.8%, and 97.1% with two, three, or four lacO direct substitutions, respectively. The loss of expression may be due to the disruption of promoter structures by direct substitution, which is a common problem in inducible promoter design[42]. In contrast, the DeepSEED-designed promoter restored high expression levels and could even outperform the original promoters (Fig. 4c, left). The average expression level of DeepSEED-designed promoters under

induced conditions decreased with increasing lacO site numbers but showed 8.96-, 6.54-, and 47.96-fold increases compared with the corresponding substitution promoters with two, three, or four lacO sites, respectively. Meanwhile, introducing lacO sites certainly generated the ability to respond to IPTG. The fold-change of DeepSEED-designed promoters showed great improvement with an increase in the number of lacO sites (Fig. 4c, right). The average fold-change of the designed promoters with two lacO sites was lower than that of the substitution group, which was caused by the higher leaky expression level under the uninduced condition. For designed promoters with three or four lacO sites, each had on average 2.20- and 2.94-fold increases compared with the corresponding substitution promoters. The results showed that DeepSEED has the ability to design high-performance inducible promoters with different lacO sites introduced. The DeepSEED-designed promoters exhibited higher levels of leaky expression compared to the corresponding substitution sequences on average (Supplementary Fig. 16a). This can be attributed to the fact that the leaky expression was not the primary focus during the design process, given the lack of leaky expression training datasets. However, we successfully addressed this issue by introducing additional lacO sites, resulting in a significant reduction in leaky expression and enabling the generation of high-performance promoters.

We further compared the performance of synthetic promoters with two widely used IPTG-inducible promoters, placUV5[43] and pLlacO1[44] (Fig. 4d). Almost all of the substitution promoters showed low induced expression levels and thus are difficult to be employed for synthetic biology applications. In contrast, DeepSEED-designed promoters showed wide distribution with different induced expression levels and fold-changes, which could satisfy various demands for fine-tuning genetic circuits. Moreover, there was a trade-off in fold-change and induced expression levels, with different numbers of lacO sites (Fig. 4d, Supplementary Fig. 16b), suggesting that the performance of synthetic promoters could be easily adjusted by altering the number of lacO sites. We also tested a subset of DeepSEED-designed IPTG-inducible promoters in M9 and EZ-rich culture medium (Supplementary Fig. 10d, Supplementary Data 1). Remarkably, a number of the designed promoters demonstrated higher induced expression levels and fold-changes compared to the commonly-used pLlacO1 promoter.

We investigated the distribution of synthetic promoters in the functional space by embedding the outputs of the penultimate layer in the predictor model using t-SNE (Fig. 4e). We found that both backbone and substitution promoters were widely distributed in space, while DeepSEED-designed promoters tended to fall in the high-activity region. These results supported the notion that the generated flanking regions were more compatible with the motif 'seeds' and target properties.

### Design of Dox-inducible promoters in mammalian cells

We further extended the DeepSEED model to mammalian cell data to test its potential in designing eukaryotic promoters. Here, we focused on the Tet response element (TRE) promoter[45,46], one of the most widely used inducible promoters in mammalian cells. Taking the Tet-On system as an example, the transcription of the TRE promoter would be activated in the presence of the reverse tetracycline-controlled transactivator (rtTA) protein, which binds to the tetO site in the TRE promoter. When tetracycline or its analogs, such as doxycycline, are added, the rtTA would be activated, and thus, gene expression would be induced (Fig. 1d). The classic TRE promoter contains seven direct repeats of tetO sites linked by repeated flanking sequences and has been used in the Tet-On system since 1995[46]. Previous engineering approaches for the TRE system focused on altering key amino acid residues in tetracycline-controlled proteins to achieve higher fold changes between induced and repressed states[47]. However, the design of the TRE promoter sequence progressed slowly due to the lack of an effective optimization strategy.

We chose the Tet-On system as an example to validate the performance of DeepSEED (Fig. 5a). We preserved the miniCMV promoter sequence as well as the tetO sites as 'seed' sequences and then generated flanking sequences with DeepSEED. The generator model was trained on the enhancer sequences in the HEK293 cell line from the Human ACtive Enhancers to interpret Regulatory variants (HACER) dataset[48]. During the training process, we annotated all known motifs of sequences in the training dataset with the JASPAR database[49] and kept the motifs while masking their flanking sequences as model input. DeepSEED aimed to recover their masked sequences to learn the regulatory patterns within flanking sequences. To avoid the increase in non-induced baseline expression, we removed all DeepSEED-designed sequences with potential binding sites of other TFs (Supplementary Fig. 6c). The detailed sequences of the in vivo validated promoters are provided in Supplementary Data 1.

Due to the sequence length limit in the training data, we first truncated the TRE promoter with three tetO sites as the initial template for flanking sequence optimization (Fig. 5b). After examining the predicted promoter performance from the predictor model and filtering out all sequences with known motifs, we obtained 12 optimized promoters to test in vivo (Supplementary Notes). Though these DeepSEED-generated sequences were very diverse from the 3-tetO template promoter, as well as the low similarity with the Homo sapiens genome (edit distance difference 67.15 ± 2.38%, Supplementary Fig. 8b), 75% of them showed higher expression levels than the 3-tetO template, with improvements of up to 2.46-fold (Fig. 5c, Supplementary Fig. 17a). Moreover, 50% of the designed promoters showed a higher fold-change, with improvements of up to 1.41-fold. Four designed promoter sequences showed improvement in both induced activity and fold-change. It is worth noting that some designed 3-tetO promoters even showed comparable induced activity to the whole-length TRE promoter with seven tetO sites, but the length is only 54.4% of the whole length. In addition, as the flanking sequences among tetO sites were optimized to non-repeating sequences, the designed promoter might have increased the DNA stability. These shorter promoters could facilitate vector construction while maintaining comparable promoter properties.

We then integrated the flanking sequences of the well-performing 3-tetO promoters to design 7-tetO promoters (Fig. 5b). We validated eight double-combination promoters and ten triple-combination promoters by integrating two and three types of optimized 3-tetO flanking sequences, respectively, in vivo. As a result, 77.8% of designed promoters showed induced activity improvement compared with the original 7-tetO-TRE promoter (Fig. 5d, Supplementary Fig. 17b, c), with an average 1.13-fold improvement and a maximum improvement of 1.23-fold. A total of 83.3% of the DeepSEED design promoters showed a higher fold-change, with improvements of up to 1.61-fold. A total of 72.2% of the designed promoters showed improvement in both induced activity and fold-change. It should be noted that the activity of the original 7-tetO-TRE promoter is comparable to that of the very strong constitutive promoters EF1A and CAGG in mammalian cells[50]; however, DeepSEED was still able to improve promoter activity without changing the main sequence architecture. We further evaluated the function of the DeepSEED-designed 7tetO Dox-inducible promoters in the HepG2 cell line. Despite the model being trained on the HEK293 dataset, the majority of the designed promoters exhibited consistent performance in both cell lines (Supplementary Fig. 18). All the results suggested the necessity of optimizing flanking sequences in designing eukaryotic promoters and again indicated the power of DeepSEED to learn regulatory patterns within flanking sequences.

## Discussion

In this study, we proposed DeepSEED, an AI-aided flanking sequence optimization method for synthetic promoter design. DeepSEED divided the promoter design process into two steps: first setting 'seed'

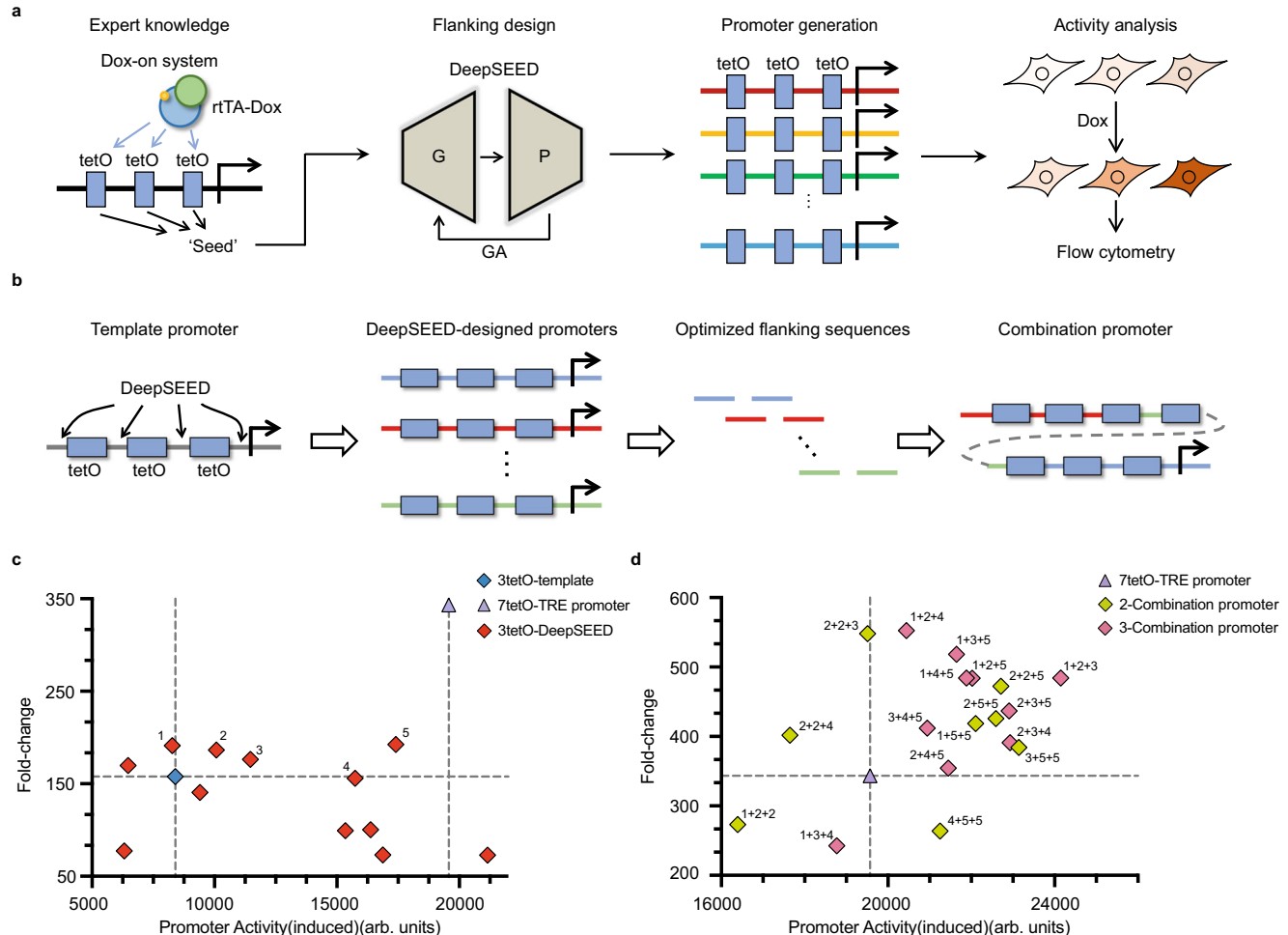

**Fig. 5 | The design of Dox-inducible promoters in HEK293 cells. a** The design process of the Dox-inducible promoters. **b** Schematic representation of the template, DeepSEED-designed, and combination promoter construction tested in vivo. **c, d** Two-dimensional images showing the induced promoter activity and fold-change of the designed promoters, 3-tetO DeepSEED designed promoters in **c**, and 7-tetO combination promoters in **d**. Numbers 1 to 5 in **c** represent selected 3-tetO promoters and they were combined to generate the 7-tetO promoters in **d**. Each dot represents the average of three biological replicates. Source data are provided as a Source Data file.

sequences as model constraints based on expert knowledge and then generating flanking sequences via a deep learning model trained on big data. We demonstrated that this co-driven knowledge-data strategy can capture the implicit patterns in flanking sequences and thus gain the ability to efficiently optimize various types of promoters with high performance. We analyzed the sequence similarity of DeepSEED-design promoters to both the template sequences (edit distance) and natural genome (BLAST search). The results showed that the DeepSEED-designed promoters exhibited a comparable difference to the randomly generated flanking sequences groups and a lower similarity with the natural genome than the promoters designed by previous works[38,39] (Supplementary Fig. 8, Supplementary Data 3). This indicates the effectiveness of our methods in designing novel synthetic promoters, rather than merely copying the original sequences.

DeepSEED can be applied to design not only constitutive promoters but also other types of promoters that have just a single or a handful of natural references. The key to this success was that we did not treat constitutive and inducible promoters as distinct objects. Instead, we proposed a probabilistic view to unify these design tasks into the same conditional optimization framework. The specific desired promoter functions were encoded by the 'seed' sequences, while DeepSEED was employed to optimize promoters as a whole to better fit the general implicit pattern preference of active promoters. Compared to previous whole sequence generation methods[22–24],

DeepSEED, by incorporating expert knowledge, could be applied to designing tasks with few training samples, such as inducible promoters, etc.

This study reinforces the importance of flanking sequences in determining promoter properties[14]. We emphasize the importance of considering a wider range of flanking sequences in promoter design, rather than solely focusing on the adjacent motifs[18,51,52]. DeepSEED successfully learned the implicit patterns of flanking sequences, and the results of saliency maps and embedding space exploration showed that the features extracted by DeepSEED are related to promoter activity. Interpretability analysis is crucial for understanding gene expression regulation[53–55]. While we have focused on the k-mer frequencies and DNA shape features learned by our model to provide partial explanations, the biological mechanisms underlying the flanking sequence regulation remain unclear. The interpretability of deep learning models remains a challenge[14,56]. With the rapid development of interpretation methods of deep learning models[56,57], it would be possible to combine deep learning and biological experiments to uncover how features of flanking sequences matter to promoter properties in an explicit way to gain new expert knowledge.

We have experimentally validated the DeepSEED-designed promoters in a plasmid system and demonstrated their functionality and effectiveness in different cell types. However, in situations where the synthetic gene expression cassette is integrated into the genome such

as in CAR-T cell therapy[58], the behavior of regulatory elements can be influenced by their genomic context, including chromatin accessibility, nucleosome arrangement, epigenetic modifications, etc[16,59–62]. Therefore, further research and validation are necessary to assess the performance of these AI-designed promoters within the genome environment.

The current version of DeepSEED is specifically designed to optimize the expression level of promoters, as there is a lack of sufficient big data training sets for other functional properties. While many of the DeepSEED-designed inducible promoters successfully achieve high induction rates by increasing the maximum induced expression level, it should be noted that some of these promoters also exhibit elevated basal expression levels, leading to compromised induction rates. To overcome these limitations, future work needs to be done to generate sufficient task-specific experimental data with high-throughput techniques, such as massively parallel reporter assays[63], DeepSEED can be further trained on these datasets to address and optimize other critical aspects of promoter engineering, including leaky expression, sequence stability, cell type specificity, etc[10]. Using such a strategy, it is also possible to further apply the DeepSEED framework to design other types of synthetic genetic elements in various organisms.

## Methods

### Bacterial strain and plasmids

All bacterial transformations and fluorescence assays were performed in the *E. coli* strain trans5α (TransGen, CD201) [F-, φ80d, lacZΔM15, Δ(lacZYA-argF), U169, endA1, recA1, hsdR17(rk-, mk +), supE44λ-, thi-1, gyrA96, relA1, phoA]. All plasmid construction was carried out using the NEBuilder HiFi DNA assembly reaction (NEB, E2621) according to standard Gibson assembly. All *E. coli* promoter constructs were cloned into the vector pTPR with a p15A origin of replication and chloramphenicol resistance. A terminator DT5 was inserted at the 5' end of the *E. coli* promoter to avoid the influence of the upstream sequence, and the insulator Riboj was inserted at the 3' end of the promoter to ensure that the same transcript was produced. The ribosomal binding site sequence BBA_B0034 was used at the 5' end of the *sfgfp* or *mrfp* gene. Double terminators BBa_B0015 were used to terminate *sfgfp* or *mrfp* gene transcription. For IPTG-inducible promoter constructs, an additional LacI protein expression cassette was inserted at the 5' end of terminator DT5 in the opposite reading direction. For the Dox-inducible two-plasmid system, pwx156 contains the reverse tetracycline-controlled transactivator (rtTA), and pwx158 contains the Dox-inducible promoter to express EYFP. The designed Dox-inducible promoters replaced the original promoter sequence for activity measurement. The detailed sequences of the plasmids used in this article are provided in Supplementary Data 2.

### Cell line and transfection

HEK293 (293-H, from Invitrogen) and HepG2 (from the Institute of Basic Medical Sciences of CAMS) cell lines were used to test the Dox-inducible system. HEK293 and HepG2 cells were cultured in Dulbecco's modified eagle medium (DMEM) with 4.5 g/L glucose (GIBCO, 11965118) supplemented with 10% FBS (GIBCO, 16000-044), 1× NEAA (GIBCO, 11140050), and 0.5× penicillin-streptomycin (Solarbio, P1400) at 37 °C and 5% $CO_2$. Lipofectamine LTX (Invitrogen, 15338100) was used for HEK293 transfection, and Lipofectamine 3000 (Invitrogen, L3000150) was used for HepG2 transfection following the manufacturer's protocol. In transfection experiments, approximately ~$1.8 \times 10^5$ cells with 1 mL culture medium were seeded into each well of 12-well plates to be ~70% confluent after being grown for ~24 h. For the observation of Dox-inducible systems, transfection with 600 ng of the plasmids carrying design promoters with expression cassette and 600 ng of the plasmids carrying the reverse tetracycline controlled transactivator (rtTA) gene were performed in the wells of a 12-well

plate. Before transfection, the culture medium containing 1 μg/ml Dox was used to induce the expression of the promoter. Cells were harvested 24 h after transfection before flow cytometry analysis.

### Assay of *E. coli* promoter strength

All of the *E. coli* promoter activity was measured based on the expression of the *sfgfp gene*. And 21 constitutive promoters measured by the *mrfp gene* were shown in Supplementary Fig. 10a and Supplementary Data 1. The strain containing the target promoter plasmid was cultured overnight (16 h) in 5 ml LB medium supplemented with 50 μg/ml chloramphenicol at 37 °C in a shaker at 220 rpm for promoter activity validation. The overnight cultures were diluted 1:100 in fresh Luria–Bertani (LB) medium supplemented with 50 μg/ml chloramphenicol in triplicate. For inducible promoters, a final concentration of 0.1 mM IPTG was added to the medium when diluted. After incubating for another 6 - 8 hours, 150 ul of culture was added to a flat-bottomed 96-well microplate (Corning 3603), and the measurements of the optical density at 600 nm (OD600) and fluorescence (relative fluorescence units [RFU]; sfGFP: excitation at 485 nm and emission at 520 nm; mRFP: excitation at 584 nm and emission at 607 nm) were repeated with the Varioskan Flash (Thermo). The background fluorescence was measured using 150 μl fresh LB medium and a strain harboring a promoterless plasmid. The strength of the promoter was defined as the average fluorescence/OD600 after subtracting background fluorescence. The process of evaluating promoter activity in M9 (Coolaber SL0060) and EZ-rich (Coolaber MK0100) culture medium was performed in the same manner as in the LB medium. All the constitutive and IPTG-inducible promoter sequences with their promoter activity in *E. coli* were shown in Supplementary Data 1.

### Flow cytometry and data analysis

Cells were trypsinized 24 h after transfection and were then centrifuged at 300 × g for 5 min at room temperature. Then, the cells were washed with phosphate-buffered saline (PBS) once and resuspended in 1x PBS in a total volume of 300 μl. Next, the cells were analyzed using LSRFortessa (BD Biosciences). The excitation lasers (Ex), emission filters (Em), and photomultiplier tube (PMT) voltage used for respective fluorescent protein measurements are as follows: TagBFP (Ex: 405 nm laser, Em: 450/50 filter, PMT: 350 V for HEK293; 300 V for HepG2). EYFP (Ex: 488 nm laser, Em: 530/30 filter, PMT: 200 V). For each sample, ~$1 \times 10^5$ cell events were collected for downstream analysis. For data analysis, raw data were filtered through the FlowAI plugin to remove bad-quality data. Cells with a TagBFP intensity between $1 \times 10^4$ and $5 \times 10^4$ (HEK293 cells) and $8 \times 10^2$ and $5 \times 10^3$ (HepG2 cells) were selected containing the proper concentration of rtTA proteins. The EYFP mean was the activity of the designed promoters. Three independent biological replicates were performed for each promoter. All the Dox-inducible promoter sequences with their promoter activity in the HEK293 and HepG2 cell lines were shown in Supplementary Data 1.

### Experimental dataset for model training

Three high-throughput datasets were used for training the DeepSEED model. A detailed description of datasets and training data generation are as follows:

**Johns.** This dataset was obtained from Johns et al.[37]. Briefly, the author defined the regulatory sequence as 165 bp upstream of the gene start codon. The regulatory library was cloned into a p15A vector and transformed into *E. coli* MG1655. RNA-seq was performed to measure the activity of each sequence. The RNA-seq dataset contained a total of 29,249 regulatory sequences from 184 prokaryotic genomes. The functional 165 bp sequences with the activity label were selected in the training process of both constitutive and inducible *E. coli* promoter design.

**HACER**. This dataset was obtained from Wang et al.[48]. The dataset contains enhancer sequences from a large number of human cell types. From the HEK293 cell line dataset, we randomly obtained 150 bp sequences in each enhancer region and finally got a total of 26,604 enhancer sequences to train the generator in the Dox-inducible promoter design task.

**Ernst**. This dataset was obtained from Ernst et al.[64]. The dataset targets 15,720 regulatory regions and tiles at 5-nucleotide resolution in two human cell types. The RNA-seq dataset was used to train the predictors in the Dox-inducible promoter design.

### Promoter design approach
The promoter design problem can be formulated from the probability perspective. The designed sequence is denoted as $s \sim p_s$, where $p_s$ represents the natural sequence distribution. The optimization aim is to maximize the joint probability of the promoter sequence $s$ and the target property $T$, as shown in Eq. (1).

Expert knowledge determines motifs related to target properties, which we note as $m$. Other regions are defined as flanking regions and can be denoted as $f$. Then, the designed sequence can be split into:

$$s = (m, f) \tag{4}$$

Therefore, our objective can be written as follows:

$$\max_{m,f} P(m, f, T) = \max_{m,f} P(m|T) P(f|m, T) P(T)$$
$$\propto \max_{m,f} P(m|T) P(f|m, T) \tag{5}$$

The first factor refers to the process of assigning 'seed' sequences compatible with the target property according to expert knowledge. The second factor represents the flanking region distribution conditioned on 'seed' sequences and the property. We first optimized the first term and then optimized the second term fixing $m$ at the optimal value.

### Objective function
The maximization of the first factor is achieved by selecting the strong motifs responsible for the desired property, such as high or inducible transcriptional activity, according to curated observations and discoveries from laborious biology experiments. These motifs were assigned with the highest confidence, resulting in defined sequences and positions. One benefit of determining unique motifs here was to simplify the objective by reducing the optimizable variables to only flanking regions $f$. We denote the assigned motifs as $m^*$; then, the objective is to maximize:

$$\max_{f} P(f|m^*, T) \tag{6}$$

Since $m^*$ has been decided in the previous section, according to the Bayes rule in probability theory:

$$\max_{f} P(f|m^*, T) = \max_{f} \frac{P(f|m^*) P(T|f, m^*)}{P(T|m^*)}$$
$$\propto \max_{f} P(f|m^*) P(T|f, m^*) \tag{7}$$

We use $p_{f|m^*}$ to denote the natural distributions of flanking regions conditioned on $m^*$. Suppose that there are $K$ kinds of flanking region combinations $f_1, \dots f_K$.

$$\max_{f} P(f|m^*) P(T|f, m^*) = \max_{i} P(f_i|m^*) P(T|f_i, m^*) \tag{8}$$

Notably, $K$ is extremely large; for example, $K = 4^{100}$ when the length of concatenated flanking regions is 100 bp. To approximate $p_{f|m^*}$, a generative neural network $G$ was trained to approximate the sample distribution, i.e., $p_g \sim p_{f|m^*}$, where

$$f_i = G(m^*, z) \tag{9}$$

where $z$ is a latent variable in the generative model.

Given the large search space, we narrow it to the output sequences of $G$. To simplify this equation, all flanking regions generated by $G$ are treated as equivalently important, with the probability of $p_0$, while the probability of other combinations is zero. Therefore,

$$\max_{i} P(f_i|m^*) P(T|f_i, m^*)$$
$$\Longleftrightarrow \max_{z} p_0 P(T|G(m^*, z), m^*) \tag{10}$$

where the term $P(T|G(m^*, z), m^*)$ is the probability of the target property given the generated sequence. We use $F$ to denote that the predictive model of properties $T$ and $F$ is trained separately. When optimizing to enhance the property, the objective function is:

$$\max_{z} F(G(m^*, z), m^*) \tag{11}$$

### DeepSEED generative and predictive networks
The flanking regions are generated through the generator $G$ while preserving predetermined motifs. The input is the concatenation of $m^*$ and $z$, and then we feed forward the input to a linear layer, and the results are reshaped into $X_m \in R^{n \times d_k}$, where $n$ represents the number of samples and $d_k$ denotes the channels. To embed the long-range dependency between regions into the network, we utilize the multi-head attention mechanism to learn the genetic element semantics in both the generator and discriminator models. The multi-head attention operation can be written as:

$$\text{MultiHead}(X_m, X_m, X_m) = \text{concat}(head_1, \dots, head_o),$$
$$head_i = \text{softmax}\left(\frac{X_m A_i{}^Q (X_m A_i{}^K)^T}{\sqrt{d_k}}\right) X_m A_i{}^V \tag{12}$$

where we denote the learnable parameters as $A_i{}^Q, A_i{}^K, A_i{}^V$. After passing one attention layer, two resblock layers with convolution operations are added to further extract features. The WGAN-GP architecture is chosen to alleviate mode collapse during the training process. Let us define the discriminator in cGANs as $D$. The loss function of cGANs can be described as follows:

$$L_{cGAN} = E_f[\log D(m, f)] + E_z[\log(1 - D(m, G(m, z)))] \tag{13}$$

Following similar tasks in image generation such as Pix2Pix[30], we add one L1 loss, which can be written as:

$$L_{L1} = E_{f,z}\left[||f - G(m, z)||\right] \tag{14}$$

The final optimizing object in implicit pattern learning is:

$$G = \arg \min_{G} \max_{D} L_{cGAN} + \lambda L_{L1}(G) \tag{15}$$

We take *E. coli* constitutive promoter design as an example. The training dataset contains 165 bp functional *E. coli* promoters with different TSSs from the library. We filtered these promoters by finding the regions from distal to TSS larger than 75 bp, finding promoters restricting −10 and −35 regions to 1–21 bp and 25–45 bp upstream of TSS with the spacer region length ranging from 10-24 bp. The input of the

DeepSEED generator is the combination of one-hot sequence encodings with determined motifs, where regions other than these motifs are set to random vectors. For the discriminator, the input is the concatenate of one-hot encodings with motifs and the generated sequences. The batch size of the training samples is 32, and we trained the model for 50,000 batches with the Adam optimization method. The learning rate, beta1, and beta2 were set to 0.0001, 0.5, and 0.9, respectively.

Accurate predictions of target properties are crucial to optimize the flanking regions. Predictions of biological properties with a deep learning model often suffer from the overfitting problem, requiring a carefully designed network architecture. The first layer of the Deep-SEED prediction model was 1d convolutional kernels with 64 output channels to capture regional information. Then, we added the long short-term memory (LSTM)[65] architecture to capture the relationships between regions. To extract potential long-range relationship factors leading to the target properties, we adopted the DenseNet[66] architecture to efficiently improve the network depth. We set 4 dense blocks with 2, 2, 4, and 2 dense layers in each block. The growth rate was set to 32. Each dense layer contains two convolutional layers, the kernel sizes of 1 and 3. Finally, we added one fully connected layer to predict the property of interest (Supplementary Notes).

## Optimization through the genetic algorithm

The objective function is shown in Eq. (11), as mentioned above. We utilized the genetic algorithm (GA) to optimize the objective function. The GA[67] module in the sko Python package was adopted to implement the genetic algorithm. The optimization process using the genetic algorithm from step $t$ to $t+1$ can be written as:

$$z^{t+1} = GA(z^t, T, G) \tag{16}$$

We set the number of seeds in GA to 5*1024 with a mutation probability of 0.005. We set the epochs of GA optimization to 100 for each task. We used the "vectorization" mode in the sko package to facilitate optimization.

## DNA shape analysis

Four kinds of DNA shape, minor groove width (MGW), roll, propeller twist (ProT), and helix twist (HelT), were estimated by the prediction model[33]. In Fig. 2d, DNA shapes on the 5' end of the 'TATAAT' motif and 'TATAAAA' motif were estimated by the model, then the distributions of four kinds of DNA shapes were calculated. In Fig. 2e & 3e, the DNA shapes of high- and low-activity promoters in the training datasets, Control-1 group promoters, Control-2 group promoters, and DeepSEED group promoters were calculated. Then, four kinds of DNA shapes of each promoter were combined to form a DNA shape feature vector. All the feature vectors were encoded by the unsupervised learning model Deepinfomax[68], and then the encoded features were projected into 2-dimensional space by the dimension reduction algorithm t-SNE.

## Second promoter finding

To prevent the model from generating the second promoter in a 165 bp sequence, we checked the RNA polymerase binding possibility in a sequence. To be more specific, the −10 motif, −35 motif, and their spacing length were taken into consideration as 'promoter structure'. The −10 motif and −35 motif were combined with a 17 bp spacer sequence, and then the sequence was scanned with Find Individual Motif Occurrences (FIMO)[69] to check whether the combined motif appeared in the sequence. The p-value of the combined motif matched in the position of the sequences is shown in Supplementary Fig. 11.

## Saliency map analysis

We calculated the saliency map features[31] of 2000 functional sequences by using the predictor model. The saliency map of each sequence represents the contribution of each nucleotide to the predicted activity. Then, the 2000 saliency maps were clustered into seven clusters by k-means. Last, 30 sequences in each cluster were randomly selected to represent the cluster (Fig. 2a, Supplementary Fig. 3).

## Semantic sequence space

The semantic sequence space was defined as the space representing the semantic information directly obtained from the DNA sequence. A total of 2000 natural promoter sequences in *E. coli*, 2000 promoter sequences generated by DeepSEED, and 14,700 promoter sequences from 147 different prokaryotic species[37] were used to define the semantic sequence space. All the sequences were first encoded by Deepinfomax[68], and then the encoded features were projected into 2-dimensional space by the dimension reduction algorithm t-SNE (Supplementary Fig. 5a).

## Sequence similar diversity

The sequence similar diversity is defined as the edit distance distribution between sequences from different sequence groups. 60 random sequences, 60 natural sequences, and 60 cGAN-generated sequences in *E. coli* were selected. The random sequence similar diversity was calculated by the edit distance between each sequence in the random sequence group and each sequence in the natural sequence group as well as the cGAN-generated sequence group. Then all the calculated edit distances formed the edit distance distribution, which could be written as:

$$P_r = \left\{ ed\left(seq_i, seq_j\right), i \in r, j \in g \ or \ n \right\} \tag{17}$$

Where $P_r$ represents the edit distance distribution of the random sequence group, ed represents the edit distance calculation, $r$ represents sequences in the random sequence group, $g$ represents sequences in the generated sequence group and $n$ represents sequences in the natural sequence group. The natural and cGAN-generated sequence similar diversity were calculated in the same way (Supplementary Fig. 5b).

## Edit distance analysis

We analyzed the edit distance between the flanking regions of experimentally validated sequences and the template sequences in our study. Specifically, for *E. coli* constitutive promoters, we utilized the short J23119/J23118/J23114 promoters as respective template sequences. In the case of IPTG-inducible promoters, we employed the sequences within the substitution group as the template sequences for each group. For dox-inducible promoters, the original 3-tetO functional sequences were used as the template. It is important to note that the percentage of difference was calculated solely based on the length of the flanking sequences. The difference of edit distance between experimental sequences and template sequences could be:

$$s = \frac{ed\left(seq_{exp}, seq_{tem}\right)}{flanking(seq)} \tag{18}$$

Where $s$ represents the difference scores, ed represents the edit distance between two input DNA sequences, $seq_{exp}$ represents experimental sequences, and $seq_{tem}$ represents template sequences. flanking($seq$) represents the length of flanking regions in the $seq$. All edit distances, percentage differences and sequences can be found in Supplementary Data 3.

## BLAST search

We performed a BLAST search to compare all the experimentally validated sequences. The *E. coli* promoter sequences were compared against the *E. coli* K12 genome (taxid: 83333), while the Dox-inducible promoter sequences were compared against the Homo sapiens genome (taxid: 9606). The blastn algorithm was used with default settings

(comparing somewhat similar sequences), except for setting the e-value expect threshold to 1000 to allow for broad comparison. To establish control groups, we utilized promoters designed by Alper[38] and the majority of constitutive sigma70 promoters in iGEM BioBrick[39]. Additionally, random sequences were generated using the probability distribution {A:0.25, C:0.25, G:0.25, T:0.25}. All of the e-values and sequences are shown in Supplementary Fig. 8 and Supplementary Data 3.

## Statistics and reproducibility
No statistical method was used to predetermine the sample size. No data were excluded from the analyses. The experiments were not randomized. The Investigators were not blinded to allocation during experiments and outcome assessment. More information was provided in the Reporting Summary file.

## Reporting summary
Further information on research design is available in the Nature Portfolio Reporting Summary linked to this article.

## Data availability
The promoter sequences generated and tested in this study are available in the Supplementary Data 1 and 3. The plasmid sequences used in this study were provided in the Supplementary Data 2. Source data are provided in this paper. The raw data files for flow cytometry analysis were deposited to the Zenodo repository and are available at "zenodo.8307150". This work utilized several published datasets. Datasets proposed by Ernst et al. were used to construct the training set of the predictor in eukaryotic promoter design "GSE71279". Enhancer datasets of the HEK293 cell line called HACER were used to construct the training set of the generator in eukaryotic promoter design [http://bioinfo.vanderbilt.edu/AE/HACER/]. Potential promoters in bacteria proposed by Johns et al. were used to construct the predictor and generator in prokaryotic promoter design [https://static-content.springer.com/esm/art%3A10.1038%2Fnmeth.4633/MediaObjects/41592_2018_BFnmeth4633_MOESM4_ESM.xlsx]. The motif sequences in JASPAR database were used to find the potential binding sites in flanking sequences [https://jaspar.genereg.net/]. Source data are provided in this paper.

## Code availability
The computer source code is available from the public GitHub repository at https://github.com/WangLabTHU/deepseed. The source code was deposited to the Zenodo repository and is available at https://doi.org/10.5281/zenodo.8307150[70].

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

## Acknowledgements

This work was supported by the National Key R&D Program of China (No. 2020YFA0906900), the National Natural Science Foundation of China (Nos. 62250007, 62225307, 61721003), and a grant from the Guoqiang Institute, Tsinghua University (2021GQG1023).

## Author contributions

X.W.W., P.C.Z., H.C.W., and H.W.X. conceived the study. X.W.W., H.C.W., H.W.X., and Z.R.H. implemented in silico designs. X.W.W., H.C.W., and H.W.X. performed the computational analysis. X.W.W., P.C.Z., L.Y.L., and L.W. designed the experimental analysis. P.C.Z. performed the experiments and analyzed the experimental data. X.W.W., P.C.Z., H.C.W., and H.W.X. wrote the manuscript.

## Competing interests

Tsinghua University has filed patent applications on behalf of X.W.W., H.C.W., H.W.X., P.C.Z., and W.L., pertaining to the AI-aided promoter design framework DeepSEED. The remaining authors declare no competing interests.
