## [Peer Review File · Nature Communications]

REVIEWER COMMENTS

Reviewer #1 (Remarks to the Author):

I have had the opportunity to review the article "Deep flanking sequence engineering for efficient promoter design" for potential publication in Nature Communications. I am pleased to provide my review of the work, which offers an exciting contribution to synthetic biology. The authors aim to address promoter design challenges by introducing DeepSEED, a framework combining expert knowledge and deep generative neural networks. The work highlights the significance of designing promoters with desirable properties in synthetic biology. The authors argue that while human experts excel at identifying strong explicit patterns in small samples, deep learning models have proven effective in detecting weak implicit patterns in large datasets. Integrating these two approaches, DeepSEED demonstrates a clever solution for designing synthetic promoters with constraints.

An important aspect of the study is its focus on the often-neglected flanking sequences of cis-regulatory elements. The authors suggest that these sequences have been historically overlooked and arbitrarily determined in promoter design. DeepSEED attempts to overcome this limitation by capturing implicit features in flanking sequences, such as k-mer frequencies and DNA shape features, which are important for determining promoter properties to some extent.

The authors present experimental results demonstrating DeepSEED's potential efficacy in improving the expression of Escherichia coli constitutive, IPTG-inducible, and mammalian cell doxycycline (Dox)-inducible promoters.

The manuscript is generally well-written, clearly explaining the methodology, experimental setup, and results. The authors make an effort to make the article accessible to a wide range of readers, including those outside the immediate field of synthetic biology. The code repository has good instructions on installing and using their approach.

In summary, I recommend further consideration of this article for publication in Nature Communications.

Here are a few specific minor comments to address:

- 1) it would be great if you could show how much different (edit distance, % of sequence) the experimentally validated DeepSEED sequences are from the closest natural sequences.

2) The author could also consider providing more information on the potential limitations of the DeepSEED approach and avenues for future research. This could help readers better understand the scope of the work and its implications for the field.

3) This predictor achieved a high Pearson correlation coefficient (PCC) of 0.74 compared with our previous CNN-based predictive model¹⁷ (PCC = 0.65) in the *E. coli* promoter training dataset (Supplementary Fig. S5).

The differences are not dramatic, as these can have many reasons because of how two models were trained, how hyperparameters are optimised and how much effort is put into tuning these. I suggest removing it as this is not a proper formal benchmark, as CNNs for these types of tasks are also known to be very good.

4) "designed sequences correlated well 125 (K = 4 to 6) at the global scale (Fig. 2c, Supplementary Fig. S3a), which was much better than

the sequences designed by our previous Wasserstein generative adversarial network–gradient penalty (WGAN-GP) model¹⁷ (Supplementary Fig. S3d).

Fix scientific language or remove it; "results much better" does not say much.

5) I could not find how datasets were split into training/validation/test in methods or Supplementary materials and how hyperparameters were optimised; this should be added and described as this is essential.

6) I believe this reference is highly related to the proposed approach and deserves citation:

<https://www.nature.com/articles/s41467-022-32818-8#Bib1>

Reviewer #2 (Remarks to the Author):

The transcription factor binding sites flanking sequences have significant influence on promoter properties. However, due to the limited understanding the molecular mechanism, it is difficult to summarize the feature into explicit promoter design rules. To overcome this issue, the authors developed deep learning models to predict and generate the flanking sequences to fit into the implicit patterns of promoters. This AI-assisted method was used to optimize the prokaryotic constitutive

promoters, prokaryotic IPTG-inducible promoters and eukaryotic doxycycline-inducible promoters, all of cases showed significant improvement in achieving desired promoter properties with high success rates. Generally, this manuscript is well-structured and the developed method serves as a powerful AI-aided tool for designing synthetic promoters. I suggest a revision before the work can be accepted.

1. Compared with previously constructed deep learning model, the advantages of the DeepSEED developed in this study should be highlighted in the text.
2. What is the reason for the AI generated promoter sequences are longer than the initial promoter sequence should be explained.
3. Except transcription activity, the leaky expression is another important promoter property. For an inducible promoter, high transcription rate usually causes high leaky expression. Thus, the leaky expression activity of the developed promoters should also be characterized.
4. When using this model to generate new inducible promoters, the authors characterized the fold changes of the promoter transcription activity. If the developed promoter has lower leaky expression, it will show a higher fold change. It is necessary to discuss whether the dramatically increased fold change is resulted from the decreased leaky expression.
5. The number of significant digits reported is excessive in Line 305.

Reviewer #3 (Remarks to the Author):

Zhang et al. describe an AI-aided approach to optimize both constitutive and inducible promoter sequences in prokaryotic and eukaryotic cells. By using previous knowledge to train their model and also considering flanking sequences they manage to design both constitutive and inducible promoters' sequences with strong activity. This study describes a novel implementation of deep learning techniques to improve promoters' properties. It could be of broad relevance to synthetic circuit designs in bioengineering.

I acknowledge the high-quality technical work carried out by the authors of the manuscript. The use of deep learning to improve de novo promoters' designs is a much-needed but not a new idea. While I am excited by the more accurate models to produce promoters with high strength and better fold change, I hesitate to recommend the work for publication in this journal for the following major reason.

Major comments:

- The limited scope of the impact. It is unclear to me that the results described in this study are going to be of broad interest to the nature communications community. Because there is no novel molecular or mechanistic insight described in this study. It is unclear if the promoter's design will work with the expression of different genes or if the growth conditions change. It is also unclear if the model captures the main biophysical properties of promoters as they were tested in only one condition.

- Limited advancement in the approach, and no new insights. The use of deep learning for promoter design is compelling. But there is no real advancement in the underlying approach: the data set is not new, the algorithm is standard, and the modifications apply to promoters' sequences are classic. The authors specify that the work provides precious information about the importance of the impact of the flanking regions on promoter activity, but little new information is provided here that was not previously shown in the literature.

- Minor improvement of promoters' properties. The authors mainly improve constitutive promoters in *E. coli* to a maximal expression level similar to J23119. The fold change of inducible promoters is better but in the same range as existing systems and with strong variability in the data.

Minor comments:

- The supplementary figures are randomly ordered in the text

- In the main text, it would be better to have a reference to the specific section of the methods and not just methods

- Line 32: can you develop the impact of promoter design on gene therapy?

- Line 41: this sentence is not clear, you should rephrase it. "dependency between TFBS and the flanking region" is too vague. How is a "low-affinity binding site" resulting from the flanking region, "non-canonical DNA structures" different from physiochemical properties of DNA shapes?

- Line 69: the panel of each figures are cited in random order in the text. You should start with panel a then b, etc...

- Line 73: define local and global promoter regions.

- Line 73: I don't understand how these results showed the vital importance of the flanking region. Can you develop this part?

We are grateful for the valuable feedback and suggestions from the three reviewers and have substantially revised the manuscript. We provided more precise and detailed descriptions in the manuscript to offer a comprehensive understanding of our work and performed additional experiments to test the function of the synthetic promoters. Most notably: 1) We performed the edit distance analysis and a standard nucleotide BLAST search to demonstrate that DeepSEED is capable of generating novel synthetic promoters, rather than simply copying the original sequences; 2) We provided detailed results on the leaky expression of inducible promoters and offered explanations related to the fold-changes of these promoters to illustrate the approach taken by DeepSEED in designing promoters; 3) We performed additional experimental tests using another reporter gene *mrfp*, in two other types of culture medium (EZ-rich and M9), and in the human HepG2 cell lines, to further validate the robustness of DeepSEED-generated promoters.

We are confident that these improvements, along with other suggested changes from the reviewers, have significantly enhanced the manuscript's impact and made the results more convincing and understandable. Specific responses to reviewer comments are noted below in blue.

REVIEWER COMMENTS

Reviewer #1 (Remarks to the Author):

I have had the opportunity to review the article "Deep flanking sequence engineering for efficient promoter design" for potential publication in Nature Communications. I am pleased to provide my review of the work, which offers an exciting contribution to synthetic biology. The authors aim to address promoter design challenges by introducing DeepSEED, a framework combining expert knowledge and deep generative neural networks. The work highlights the significance of designing promoters with desirable properties in synthetic biology. The authors argue that while human experts excel at identifying strong explicit patterns in small samples, deep learning models have proven effective in detecting weak implicit patterns in large datasets. Integrating these two approaches, DeepSEED demonstrates a clever solution for designing synthetic promoters with constraints.

An important aspect of the study is its focus on the often-neglected flanking sequences of cis-regulatory elements. The authors suggest that these sequences have been historically overlooked and arbitrarily determined in promoter design. DeepSEED attempts to overcome this limitation by capturing implicit features in flanking sequences, such as k-mer frequencies and DNA shape features, which are important for determining promoter properties to some extent.

The authors present experimental results demonstrating DeepSEED's potential efficacy in improving the expression of *Escherichia coli* constitutive, IPTG-inducible,

and mammalian cell doxycycline (Dox)-inducible promoters.

The manuscript is generally well-written, clearly explaining the methodology, experimental setup, and results. The authors make an effort to make the article accessible to a wide range of readers, including those outside the immediate field of synthetic biology. The code repository has good instructions on installing and using their approach.

In summary, I recommend further consideration of this article for publication in Nature Communications.

Here are a few specific minor comments to address:

1) it would be great if you could show how much different (edit distance, % of sequence) the experimentally validated DeepSEED sequences are from the closest natural sequences.

We thank Reviewer 1 for bringing the issue to our attention. In response, we analyzed the edit distance between the experimentally validated DeepSEED sequences and the initial sequences in our study. The results revealed significant differences between the DeepSEED-generated promoters and the initial promoter sequences.

Specifically, in the design of *E. coli* constitutive promoters, the DeepSEED-generated promoters exhibited sequence differences of $51.39 \pm 4.66\%$, $45.14 \pm 3.63\%$, and $46.06 \pm 5.23\%$ compared to the J23114, J23118, and J23119 initial sequences, respectively. These results were found to be comparable to the Control-2 group, where randomly generated flanking sequences exhibited differences of $52.31 \pm 6.33\%$, $51.39 \pm 3.55\%$, and $50.93 \pm 6.42\%$ from the J23114, J23118, and J23119 initial sequences, respectively. Regarding the design of *E. coli* inducible promoters, the 2-lacO, 3-lacO, and 4-lacO promoters exhibited differences of $58.33 \pm 4.08\%$, $60.11 \pm 4.00\%$, and $61.84 \pm 5.39\%$, respectively, when compared to the template sequences. For the doxycycline-inducible promoter, the 3-tetO DeepSEED-generated sequences had a difference of $67.15 \pm 2.38\%$ compared to the 3-tetO template promoter. The details of the edit distance analysis can be found in the "Methods" section and Supplementary Table 3.

Furthermore, a standard nucleotide BLAST search on experimentally tested DeepSEED promoters was conducted against the natural *E. coli* genome (taxid:83333) or Homo sapiens (taxid:9606), and no high-similarity matches were found (Supplementary Table 3, see the "Methods" section). The e-value of the DeepSEED-designed promoter and random sequence is at the same level and shows lower similarity with the natural genome than the promoters designed by Alper et al (Alper et al., 2005) and most constitutive promoters from iGEM BioBrick standard parts (Smolke, 2009) (Supplementary Fig. 8).

2) The author could also consider providing more information on the potential limitations of the DeepSEED approach and avenues for future research. This could help readers better understand the scope of the work and its implications for the field.

We appreciate Reviewer 1 for the suggestion. We have expanded the “Discussion” section to include a more detailed discussion on the limitations of the DeepSEED approach and potential avenues for future research.

Firstly, we acknowledge that the current version of DeepSEED is specifically designed to optimize the activity of promoters, as there is a lack of sufficient big data training sets for other functional properties. Thus, for the inducible promoter design task, while most of the DeepSEED-designed promoters successfully achieve high induction rates by increasing the maximum induced expression level, it should be noted that some IPTG-inducible promoters also exhibit elevated basal expression levels, leading to compromised induction rates. To overcome these limitations, future works need to generate sufficient task-specific experimental data with high-throughput techniques, such as massively parallel reporter assays. DeepSEED could be further trained on these task-specific datasets, to optimize other critical aspects of promoter engineering, including leaky expression, sequence stability, cell type specificity, etc.

Secondly, we have experimentally validated the DeepSEED-designed promoters in a plasmid system and demonstrated their functionality and effectiveness in different cell types. However, in situations where the gene expression cassette is integrated into the cell genome, such as in CAR-T cell therapy (Hong et al., 2020). It is important to consider that the behavior of regulatory elements can be influenced by their genomic context, including chromatin accessibility, nucleosome arrangement, epigenetic modifications, etc. Therefore, further research and validation are necessary to assess the performance of these AI-designed promoters within the genome environment.

Thirdly, interpretability analysis is crucial for understanding gene expression regulation. While the k-mer frequencies and DNA shape features learned by DeepSEED could provide partial explanations, the mechanism underlying the flanking sequences regulation remains unclear. Further exploration of the interpretability of deep learning models may enhance our understanding. The weak but important regulation in the flanking regions, along with the interaction between flanking regions and adjacent motifs are likely to ignite widespread interest in the explanation of transcription regulation.

3) This predictor achieved a high Pearson correlation coefficient (PCC) of 0.74 compared with our previous CNN-based predictive model¹⁷ (PCC = 0.65) in the E.

coli promoter training dataset (Supplementary Fig. S5).

The differences are not dramatic, as these can have many reasons because of how two models were trained, how hyperparameters are optimised and how much effort is put into tuning these. I suggest removing it as this is not a proper formal benchmark, as CNNs for these types of tasks are also known to be very good.

We appreciate this suggestion. In response to this feedback, we have relocated the detailed model performance comparison to the part “Train/Test/Validation Set Splitting and DeepSEED performance comparison” in the Supplementary Notes.

4) "designed sequences correlated well 125 (K = 4 to 6) at the global scale (Fig. 2c, Supplementary Fig. S3a), which was much better than the sequences designed by our previous Wasserstein generative adversarial network–gradient penalty (WGAN-GP) model17 (Supplementary Fig. S3d). Fix scientific language or remove it; "results much better" does not say much.

We are grateful to Reviewer 1 for bringing up this point. According to the suggestion, we have moved the comparison of model performance to the Supplementary Notes and revised the description to provide a more accurate representation. This adjustment allows us to better focus the main manuscript on our proposed method and its key contributions.

5) I could not find how datasets were split into training/validation/test in methods or Supplementary materials and how hyperparameters were optimised; this should be added and described as this is essential.

We appreciate Reviewer 1 for raising the concern regarding the training/validation/test group of the model. We have now added an explanation in the Supplementary Notes regarding the training/validation/test set splitting. To develop the prediction model, we partitioned the samples into three sets: 80% for the training set, 10% for the validation set, and 10% for the test set. Regarding the training of the generator, as it involves unsupervised learning, we did not apply specific dataset partitioning. Instead, we monitored the k-mer frequencies during the training process. These monitoring steps are included in our model code, and we have provided more detailed descriptions regarding this in the Supplementary Notes.

6) I believe this reference is highly related to the proposed approach and deserves citation:

<https://www.nature.com/articles/s41467-022-32818-8#Bib1>

In response to Reviewer 1's suggestion, we have revised the “Introduction” to incorporate similar works that have previously explored deep learning methods for regulatory element design and highlighted the unique aspects of our method.

Reviewer #2 (Remarks to the Author):

The transcription factor binding sites flanking sequences have significant influence on promoter properties. However, due to the limited understanding the molecular mechanism, it is difficult to summarize the feature into explicit promoter design rules. To overcome this issue, the authors developed deep learning models to predict and generate the flanking sequences to fit into the implicit patterns of promoters. This AI-assisted method was used to optimize the prokaryotic constitutive promoters, prokaryotic IPTG-inducible promoters and eukaryotic doxycycline-inducible promoters, all of cases showed significant improvement in achieving desired promoter properties with high success rates. Generally, this manuscript is well-structured and the developed method serves as a powerful AI-aided tool for designing synthetic promoters. I suggest a revision before the work can be accepted.

1. Compared with previously constructed deep learning model, the advantages of the DeepSEED developed in this study should be highlighted in the text.

We thank Reviewer 2 for bringing this issue to our attention. We have included an additional explanation in the "Introduction" section and reorganized the sentences in the "Discussion" section to better emphasize the abilities and advantages of DeepSEED.

On the one hand, previous studies have primarily focused on whole sequence generation methods for designing constitutive promoters (Kotopka & Smolke, 2020; Y. Wang et al., 2020; Zrimec et al., 2022). However, these data-driven deep learning models face limitations when it comes to designing promoters with specific properties, such as inducible or tissue-specific promoters. This is mainly due to the scarcity of available promoter datasets for training. In contrast, we take advantage of the strong features identified by humans as priors and combine them with a deep learning framework to provide a general tool to design multi-types of promoters. On the other hand, the performance of DeepSEED was improved by well-designed network structures and training strategies. We incorporated attention-based layers in cGAN to capture the widespread long-range interactions in regulatory codes and utilized a DenseNet-LSTM-based predictor to enhance activity prediction performance. We compared the constitutive promoters designed by DeepSEED with our previous whole sequence generation methods (Y. Wang et al., 2020) and observed a 6.73-fold improvement in promoter activity (Fig. 3d).

2. What is the reason for the AI generated promoter sequences are longer than the initial promoter sequence should be explained.

We appreciate Reviewer 2 for raising concerns about sequence length during the design process. The primary reason for selecting a longer sequence length was due

to the sequence length of the large-scale MPRA dataset used for model training (Johns et al., 2018). Accurate and diverse gene expression data, along with sufficient sequence diversity, are essential for training deep learning models to learn transcription regulation. The dataset (Johns et al., 2018) was used for its comprehensive collection of 29,249 regulatory sequences with accurately measured expression data for 165 bp length sequences using massively parallel reporter assays in *E. coli*. To maximize the performance of the predictor, we aimed to edit *E. coli* sequences to be 165 bp in length, consistent with the aforementioned dataset. And a length of 165 bp is considered sufficient for *E. coli* promoter sequences (Mitchell et al., 2003).

In fact, DeepSEED is flexible and capable of generating arbitrary length sequences shorter than the training sequences. DeepSEED can easily assign the partial sequence of the backbone plasmids as the 'seed' sequences to control the length of the promoters designed by the model.

3. Except transcription activity, the leaky expression is another important promoter property. For an inducible promoter, high transcription rate usually causes high leaky expression. Thus, the leaky expression activity of the developed promoters should also be characterized.

We appreciate Reviewer 2 for raising concerns about leaky expression. Leaky expression is a crucial aspect of inducible promoters and significantly contributes to their overall function. In the previous version, we presented the induced expression level and fold-change in Fig. 4c and Fig. 5c&d. As shown in these figures, because of leaky expression, some promoters with very high induced expression only showed moderate fold-change levels. In this revision, we further included figures in Supplementary Fig. 16a and 17 that specifically show the leaky expression for IPTG-inducible and Dox-inducible promoters. Further explanations and details regarding this issue can be found in the subsequent reply and "Results" sections of the manuscript.

4. When using this model to generate new inducible promoters, the authors characterized the fold changes of the promoter transcription activity. If the developed promoter has lower leaky expression, it will show a higher fold change. It is necessary to discuss whether the dramatically increased fold change is resulted from the decreased leaky expression.

We sincerely appreciate Reviewer 2 for raising concerns regarding the issue of leaky expression. In response to this concern, we have made several additions and modifications to the manuscript. In the "Results" and "Discussion" sections, we have added the discussion related to the leaky expression of the promoter. Detailed

information and figures illustrating the leaky expression in different promoter design tasks can be found in Supplementary Fig. 16a and 17.

In the inducible promoter task, we observed that while the DeepSEED-generated promoters demonstrated improvements in fold-change and induced activity, they also exhibited higher levels of leaky expression compared to the corresponding control sequences on average. This is because the current primary objective of DeepSEED was to enhance the expression level by optimizing the flanking sequences. The leaky expression has not been optimized in the current version of DeepSEED due to the lack of training data.

To address the issue of leaky expression, we implemented different strategies for designing inducible promoters. In the IPTG-inducible promoter task, we introduced additional lacO sites at specific positions to further repress the binding of RNA polymerase (RNAP) to the promoter sequences. In the Dox-inducible promoter task, we removed promoter sequences that had potential binding sites for other transcription factors, minimizing unintended TF binding that could contribute to leaky expression.

In this revision, we have added a paragraph in the “Discussion” to further explain the limitation of the current version of DeepSEED. It does not specifically optimize for leaky expression due to the lack of task-specific training data. However, we emphasize that as more task-specific experimental data becomes available in the future, DeepSEED can be further trained on these datasets to address and optimize other aspects of promoter engineering, including leaky expression.

5. The number of significant digits reported is excessive in Line 305.

Thank you for the detailed reminder. We have made the requested revision to Line 305 (new revision in Line 349). The number of significant digits has been adjusted to ‘with an average 1.13-fold improvement and a maximum improvement of 1.23-fold.’.

Reviewer #3 (Remarks to the Author):

Zhang et al. describe an AI-aided approach to optimize both constitutive and inducible promoter sequences in prokaryotic and eukaryotic cells. By using previous knowledge to train their model and also considering flanking sequences they manage to design both constitutive and inducible promoters’ sequences with strong activity. This study describes a novel implementation of deep learning techniques to improve promoters’ properties. It could be of broad relevance to synthetic circuit designs in bioengineering.

I acknowledge the high-quality technical work carried out by the authors of the manuscript. The use of deep learning to improve de novo promoters' designs is a much-needed but not a new idea. While I am excited by the more accurate models to produce promoters with high strength and better fold change, I hesitate to recommend the work for publication in this journal for the following major reason.

Major comments:

- The limited scope of the impact. It is unclear to me that the results described in this study are going to be of broad interest to the nature communications community. Because there is no novel molecular or mechanistic insight described in this study. It is unclear if the promoter's design will work with the expression of different genes or if the growth conditions change. It is also unclear if the model captures the main biophysical properties of promoters as they were tested in only one condition.

We appreciate the suggestion provided by Reviewer 3. The key aspect of our work is the optimization of flanking sequences, resulting in significant improvements in promoter properties. While previous studies have highlighted the importance of flanking sequences, the design or optimization of these sequences has been challenging due to the absence of explicit rules and available tools. DeepSEED overcomes this challenge by capturing the implicit patterns in flanking sequences from big data and using these implicit patterns to generate functional sequences. Notably, through the optimization of flanking sequences, our designed promoters exhibited better expression and fold-change characteristics compared to the widely used template promoters.

Moreover, it is important to note that recent data-driven deep learning models have limitations when it comes to designing promoters with specific properties, such as inducible or tissue-specific promoters. This is primarily due to the scarcity of available promoters with the desired properties for use in training. In many cases, researchers have relied on massively parallel reporter assay experiments to generate large datasets for specific promoter designs. In our work, we leverage the strong features identified by human experts and combine them with a deep learning framework to provide a general tool for optimizing flanking sequences in promoter design. The design framework we propose has the potential to inspire the design of various other genetic elements.

In this revision, we performed additional tests using another reporter gene *mrfp*, in two additional types of culture medium (EZ-rich and M9), and in the human HepG2 cell lines, to further validate the robustness of DeepSEED-generated promoters. For constitutive promoters, we validated their function using mRFP as the target gene, demonstrating a high correlation (PCC = 0.83) with the sfGFP expression system (Supplementary Fig. 10a). This indicates that the promoter tested in the sfGFP expression system can be used to express other genes of interest. Furthermore, we evaluated the performance of the designed promoters in *E. coli* using M9 and EZ-

rich culture mediums with stable components (Supplementary Fig. 10b,c). The designed constitutive promoters, with varying -10 and -35 regions, consistently exhibited high promoter activity across different culture mediums. Moreover, some of the designed IPTG-inducible promoters outperformed pLlacO1 in M9 and EZ-rich medium (Supplementary Fig. 10d). Regarding the Dox-inducible promoters, we also conducted tests on the designed 7-tetO promoters in the HepG2 cell line (Supplementary Fig. 18). Although the model was trained on the HEK293 dataset, the majority of the designed promoters exhibited higher induced expression levels compared to the 3-tetO template promoter.

Overall, these experimental results demonstrate the reliability and performance of the promoters designed by DeepSEED across different expression systems and culture conditions. We demonstrated that this co-driven knowledge-data co-driven strategy enabled us to capture implicit patterns in flanking sequences and efficiently optimize various types of promoters with high performance.

- Limited advancement in the approach, and no new insights. The use of deep learning for promoter design is compelling. But there is no real advancement in the underlying approach: the data set is not new, the algorithm is standard, and the modifications apply to promoters' sequences are classic. The authors specify that the work provides precious information about the importance of the impact of the flanking regions on promoter activity, but little new information is provided here that was not previously shown in the literature.

We appreciate Reviewer 3 for raising this concern. Previous deep learning approaches for designing regulatory elements are mainly based on fully-generated models, which require extensive training data. This has posed challenges when designing inducible promoters with limited reference sequences. The design of inducible or cell type-specific promoters has been challenging due to the lack of explicit rules and available tools. In contrast, DeepSEED incorporates expert knowledge into the deep learning model, providing a general and flexible promoter design tool that enables to design the multi-types of promoters in different species.

While previous studies have recognized the importance of flanking sequences in promoters, there are no designing tools that have incorporated these properties into promoter design. The major challenge is that, unlike strong patterns of TFBS that could be described by the PWM matrix, the flanking sequence features are weak and implicit. To overcome this challenge, we employed a well-designed deep learning framework specifically focused on the design of flanking sequences. We showed that DeepSEED could automatically capture the weak but important features like k-mer frequencies and DNA shapes of flanking sequences, and could use these features to optimize designed promoters efficiently.

In addition, we have experimentally tested hundreds of DeepSEED-generated promoters under three culture conditions, using two types of reporter genes in *E. coli*, and two types of human cell lines. The DeepSEED-generated promoters exhibited a wide range of characteristics, including activity and fold-change, and demonstrated low similarity to previously used sequences. These well-characterized de novo promoters provided a usable dataset for researchers to use. These results also advance our understanding of promoter engineering in synthetic biology.

- Minor improvement of promoters' properties. The authors mainly improve constitutive promoters in *E. coli* to a maximal expression level similar to J23119. The fold change of inducible promoters is better but in the same range as existing systems and with strong variability in the data.

We sincerely appreciate Reviewer 3 for raising concerns. DeepSEED overcomes the traditional trial-and-error approach, such as random mutation, by providing an efficient tool for generating synthetic promoters with optimized flanking sequences with a high success rate. Importantly, DeepSEED is not limited to specific promoter design tasks but can be applied to diverse types of promoter design tasks. We demonstrated the effectiveness of DeepSEED as a flexible tool for promoter design in three different promoter design tasks in *E. coli* and mammalian cells.

In the constitutive promoter task, DeepSEED has proven effective in enhancing the activity of promoters with different -10 and -35 motifs as expert knowledge. We observed max improvements in the expression levels of up to 18.8%, 27.5%, and 12.7% compared to the J23119 promoter in LB, EZ rich, and M9 medium, respectively (Fig. 3c, Supplementary Fig. 10, Supplementary Table 1). It is important to note that the J23119 promoter is already a highly potent wild-type constitutive promoter in *E. coli*. Limited by the inherent constraints of gene expression control, it is challenging to design promoters based only on endogenous expression systems to achieve significant improvements compared to J23119. For relatively low expressed backbones, very high expression elevations were observed. In particular, in the J23114 initial group, the maximum improvement was up to 71.3-fold compared to the J23114 initial sequence. In the IPTG-inducible promoter task, DeepSEED successfully incorporates the number and position of lacO sites to generate promoters with different performance characteristics. We observed significant high degrees of activity loss when directly substituting lacO sites into the backbone without flanking sequence optimization (Fig. 4c). In contrast, the DeepSEED-designed promoters exhibited a wide distribution of induced expression levels and fold-changes, enabling fine-tuning of genetic circuits to meet diverse demands. Some of the designed promoters showed improvements in both low leaky expression and high induced expression levels compared to the commonly-used pLlacO1 promoter (Supplementary Fig. 10d). For the Dox-inducible promoter task, it is worth noting that the original 7-tetO-TRE promoter displayed activity comparable to very strong constitutive promoters like EF1A and CAGG in

mammalian cells (Qin et al., 2010). However, DeepSEED was still able to improve promoter activity without altering the main sequence architecture.

In the three tasks we selected, which involved existing promoters with excellent performance, the promoters designed by DeepSEED showed further improvement. The results presented in this study showcase the potential of DeepSEED as a general promoter optimization tool, particularly in addressing the challenges associated with designing promoters in scenarios with limited training data or scarce reference templates. According to the comments, in this revised version, we further emphasized the contribution of DeepSEED in the “Introduction” and “Discussion”.

Minor comments:

- The supplementary figures are randomly ordered in the text

We appreciate this suggestion. We have reordered the supplementary figures as closely as possible according to the citation order in the main text.

- In the main text, it would be better to have a reference to the specific section of the methods and not just methods

Thank you for the suggestion. We provided a detailed description of the citation methods in the main text. Such as <see “Edit distance analysis” in “Methods”>.

- Line 32: can you develop the impact of promoter design on gene therapy?

We appreciate this suggestion and apologize for the previous unclear presentation. In response to the feedback, we have reorganized the sentences in the “Introduction” section to better explain the impact of promoter design on gene therapy.

Cao et al. have pointed out that one of the major challenges in current gene therapy approaches is maximizing the potency of the treatment (Cao et al., 2021; Joshi et al., 2019). Additionally, D. Wang et al. have emphasized the importance of using a strong and ubiquitous promoter to achieve high transgene expression has been emphasized (D. Wang et al., 2019). While some strong promoters, such as EF1 α , CMV, and CBA promoters, have been applied in transgene expression systems (Armbruster et al., 2016; Chira et al., 2015; Veron et al., 2009), they also face certain disadvantages in the variable cellular environment. These disadvantages include lower expression levels (Cao et al., 2021; Hu et al., 2014; Pacak et al., 2008), unstable activity in different cell lines (Xia et al., 2006), susceptibility to methylation inactivation (Brooks et al., 2004; Stein et al., 2010), and the size of

promoters (Domenger & Grimm, 2019; D. Wang et al., 2019). Therefore, it is essential to tailor the design of promoters to suit the specific requirements of gene therapy.

- Line 41: this sentence is not clear, you should rephrase it. "dependency between TFBS and the flanking region" is too vague. How is a "low-affinity binding site" resulting from the flanking region, "non-canonical DNA structures" different from physiochemical properties of DNA shapes?

Thank you for the suggestion. We have revised these sentences to improve clarity. Previous studies have reported the presence of potential low-affinity binding sites in the flanking regions, which can enhance TF binding. Non-canonical DNA structures refer to non-B DNA structures such as Z-DNA, G-quadruplexes, slipped structures, triple-stranded DNA structures, etc. On the other hand, DNA shape features, including Minor Groove Width (MGW), Propeller Twist (ProT), Roll, and Helix Twist (HelT), are determined by the sequence context of the corresponding base pair within a pentamer model.

- Line 69: the panel of each figures are cited in random order in the text. You should start with panel a then b, etc...

We thank Reviewer 3 for bringing this issue to our attention. We have reordered the panels in each figure as closely as possible according to the citation order in the main text.

- Line 73: define local and global promoter regions.

Thank you for the suggestion. We have revised the description of "local and global promoter regions" as "specific and entire promoter regions" that make it easier for readers to understand.

The term "local promoter region" refers to specific portions of the promoter sequence, such as the distal and proximal ends of the promoter or the left side of the -10 element. These regions were analyzed to investigate the location-specific information, such as k-mer frequency patterns difference in distal and proximal and DNA shape features in the left side of different types of -10 elements. On the other hand, the "global promoter region" encompasses the entire promoter sequence. We analyzed k-mer frequency in both natural sequences and DeepSEED-designed promoters, and investigated how DNA shape features in the full-length promoter sequences correlate with promoter activity.

By distinguishing between specific and entire promoter regions, we aimed to capture the distinct information derived from each type of sequence and provide a comprehensive analysis of promoter properties. In the revised main text, we have provided a more detailed description of the local and global promoter regions.

- Line 73: I don't understand how these results showed the vital importance of the flanking region. Can you develop this part?

We appreciate the valuable suggestion. We have reorganized the sentences to better emphasize the importance of optimizing the flanking region.

In the article, we extensively demonstrate the importance of flanking sequences in various aspects of promoter design. Firstly, we investigated the role of flanking sequences in predicting promoter activity *in silico*. We used a predictor model to predict the expression levels of 2000 functional *E. coli* promoters and analyzed the saliency maps, which revealed distinct patterns of influence from the flanking regions (Fig. 2a and Supplementary Fig. 3). Some regions exhibited similar levels of influence on expression as the well-known -10 and -35 elements.

In the constitutive promoter design task, we constructed experiments using the Control-1 group, where the flanking sequences of short J23 promoters were randomized (Fig. 3b,c). The results demonstrated that the flanking sequence surrounding the complete short J23 promoters significantly influences promoter activity. Furthermore, DeepSEED effectively improved promoter activity by optimizing the flanking sequence. In the IPTG-inducible promoter design task, we created Substitution groups by replacing specific sequences of the backbone with lacO sites (Fig. 4b,c). We observed varying degrees of activity loss in the direct substitution of sequences under induced conditions. This highlighted the crucial role of the flanking region in promoter activity. For the Dox-inducible promoter design task, DeepSEED successfully enhanced promoter activity and fold-change without altering the main sequence architecture (Fig. 5c,d).

Overall, these results consistently emphasized the necessity of optimizing flanking sequences in promoter design and further demonstrated the capability of DeepSEED to learn regulatory patterns within these flanking sequences.

Reference

- Alper, H., Fischer, C., Nevoigt, E., & Stephanopoulos, G. (2005). Tuning genetic control through promoter engineering. *Proceedings of the National Academy of Sciences of the United States of America*, *102*(36), 12678–12683.
- Armbruster, N., Lattanzi, A., Jeavons, M., Van Wittenberghe, L., Gjata, B., Marais, T., Martin, S., Vignaud, A., Voit, T., Mavilio, F., Barkats, M., & Buj-Bello, A. (2016). Efficacy and biodistribution analysis of intracerebroventricular administration of an optimized scAAV9-SMN1 vector in a mouse model of spinal muscular atrophy. *Molecular Therapy. Methods & Clinical Development*, *3*(16060), 16060.
- Brooks, A. R., Harkins, R. N., Wang, P., Qian, H. S., Liu, P., & Rubanyi, G. M. (2004). Transcriptional silencing is associated with extensive methylation of the CMV promoter following adenoviral gene delivery to muscle. *The Journal of Gene Medicine*, *6*(4), 395–404.
- Cao, J., Novoa, E. M., Zhang, Z., Chen, W. C. W., Liu, D., Choi, G. C. G., Wong, A. S. L., Wehrspau, C., Kellis, M., & Lu, T. K. (2021). High-throughput 5' UTR engineering for enhanced protein production in non-viral gene therapies. *Nature Communications*, *12*(1), 4138.

- Chira, S., Jackson, C. S., Oprea, I., Ozturk, F., Pepper, M. S., Diaconu, I., Braicu, C., Raduly, L.-Z., Calin, G. A., & Berindan-Neagoe, I. (2015). Progresses towards safe and efficient gene therapy vectors. *Oncotarget*, *6*(31), 30675–30703.
- Domenger, C., & Grimm, D. (2019). Next-generation AAV vectors-do not judge a virus (only) by its cover. *Human Molecular Genetics*, *28*(R1), R3–R14.
- Hong, M., Clubb, J. D., & Chen, Y. Y. (2020). Engineering CAR-T cells for next-generation cancer therapy. *Cancer Cell*, *38*(4), 473–488.
- Hu, C., Kasten, J., Park, H., Bhargava, R., Tai, D. S., Grody, W. W., Nguyen, Q. G., Hauschka, S. D., Cederbaum, S. D., & Lipshutz, G. S. (2014). Myocyte-mediated arginase expression controls hyperargininemia but not hyperammonemia in arginase-deficient mice. *Molecular Therapy: The Journal of the American Society of Gene Therapy*, *22*(10), 1792–1802.
- Johns, N. I., Gomes, A. L. C., Yim, S. S., Yang, A., Blazejewski, T., Smillie, C. S., Smith, M. B., Alm, E. J., Kosuri, S., & Wang, H. H. (2018). Metagenomic mining of regulatory elements enables programmable species-selective gene expression. *Nature Methods*, *15*(5), 323–329.
- Joshi, P. R. H., Cervera, L., Ahmed, I., Kondratov, O., Zolotukhin, S., Schrag, J., Chahal, P. S., & Kamen, A. A. (2019). Achieving high-yield production of functional AAV5 gene delivery vectors via fedbatch in an insect cell-one

Baculovirus system. *Molecular Therapy. Methods & Clinical Development*, 13, 279–289.

Kotopka, B. J., & Smolke, C. D. (2020). Model-driven generation of artificial yeast promoters. *Nature Communications*, 11(1), 2113.

Mitchell, J. E., Zheng, D., Busby, S. J. W., & Minchin, S. D. (2003). Identification and analysis of “extended -10” promoters in *Escherichia coli*. *Nucleic Acids Research*, 31(16), 4689–4695.

Pacak, C. A., Sakai, Y., Thattaliyath, B. D., Mah, C. S., & Byrne, B. J. (2008). Tissue specific promoters improve specificity of AAV9 mediated transgene expression following intra-vascular gene delivery in neonatal mice. *Genetic Vaccines and Therapy*, 6(1), 13.

Qin, J. Y., Zhang, L., Clift, K. L., Hular, I., Xiang, A. P., Ren, B.-Z., & Lahn, B. T. (2010). Systematic comparison of constitutive promoters and the doxycycline-inducible promoter. *PloS One*, 5(5), e10611.

Smolke, C. D. (2009). Building outside of the box: iGEM and the BioBricks foundation. *Nature Biotechnology*, 27(12), 1099–1102.

Stein, S., Ott, M. G., Schultze-Strasser, S., Jauch, A., Burwinkel, B., Kinner, A., Schmidt, M., Krämer, A., Schwäble, J., Glimm, H., Koehl, U., Preiss, C., Ball, C., Martin, H., Göhring, G., Schwarzwaelder, K., Hofmann, W.-K., Karakaya,

K., Tchatchou, S., ... Grez, M. (2010). Genomic instability and myelodysplasia with monosomy 7 consequent to EVI1 activation after gene therapy for chronic granulomatous disease. *Nature Medicine*, 16(2), 198–204.

Veron, P., Boutin, S., Martin, S., Chaperot, L., Plumas, J., Davoust, J., & Masurier, C. (2009). Highly efficient transduction of human plasmacytoid dendritic cells without phenotypic and functional maturation. *Journal of Translational Medicine*, 7(1), 10.

Wang, D., Tai, P. W. L., & Gao, G. (2019). Adeno-associated virus vector as a platform for gene therapy delivery. *Nature Reviews. Drug Discovery*, 18(5), 358–378.

Wang, Y., Wang, H., Wei, L., Li, S., Liu, L., & Wang, X. (2020). Synthetic promoter design in *Escherichia coli* based on a deep generative network. *Nucleic Acids Research*, 48(12), 6403–6412.

Xia, W., Bringmann, P., McClary, J., Jones, P. P., Manzana, W., Zhu, Y., Wang, S., Liu, Y., Harvey, S., Madlansacay, M. R., McLean, K., Rosser, M. P., MacRobbie, J., Olsen, C. L., & Cobb, R. R. (2006). High levels of protein expression using different mammalian CMV promoters in several cell lines. *Protein Expression and Purification*, 45(1), 115–124.

Zrimec, J., Fu, X., Muhammad, A. S., Skrekas, C., Jauniskis, V., Speicher, N. K., Börlin, C. S., Verendel, V., Chehreghani, M. H., Dubhashi, D., Siewers, V., David, F., Nielsen, J., & Zelezniak, A. (2022). Controlling gene expression with

deep generative design of regulatory DNA. *Nature Communications*, 13(1),
5099.

REVIEWERS' COMMENTS

Reviewer #1 (Remarks to the Author):

The authors adequately addressed previously raised comments.

Reviewer #2 (Remarks to the Author):

I think this manuscript can be accepted to be published in NC.

Reviewer #3 (Remarks to the Author):

The authors have adequately addressed my concerns.